# Self-organised segregation of bacterial chromosomal origins

**Andreas Hofmann[1†], Jarno Mäkelä[2†], David J Sherratt[2], Dieter Heermann[1], Seán M Murray[3]\***

[1]Institute for Theoretical Physics, Heidelberg University, Heidelberg, Germany; [2]Department of Biochemistry, University of Oxford, Oxford, United Kingdom; [3]Max Planck Institute for Terrestrial Microbiology, LOEWE Centre for Synthetic Microbiology (SYNMIKRO), Marburg, Germany

**Abstract** The chromosomal replication origin region (*ori*) of characterised bacteria is dynamically positioned throughout the cell cycle. In slowly growing *Escherichia coli*, *ori* is maintained at mid-cell from birth until its replication, after which newly replicated sister *ori*s move to opposite quarter positions. Here, we provide an explanation for *ori* positioning based on the self-organisation of the Structural Maintenance of Chromosomes complex, MukBEF, which forms dynamically positioned clusters on the chromosome. We propose that a non-trivial feedback between the self-organising gradient of MukBEF complexes and the *ori*s leads to accurate *ori* positioning. We find excellent agreement with quantitative experimental measurements and confirm key predictions. Specifically, we show that *ori*s exhibit biased motion towards MukBEF clusters, rather than mid-cell. Our findings suggest that MukBEF and *ori*s act together as a self-organising system in chromosome organisation-segregation and introduces protein self-organisation as an important consideration for future studies of chromosome dynamics.
DOI: https://doi.org/10.7554/eLife.46564.001

**\*For correspondence:**
sean.murray@synmikro.mpi-marburg.mpg.de

[†]These authors contributed equally to this work

**Competing interests:** The authors declare that no competing interests exist.

## Introduction

The faithful and timely segregation of genetic material is essential for all cellular life. In eukaryotes the responsibility for chromosome segregation lies with a well-understood macromolecular machine, the mitotic spindle. In contrast, the mechanisms underlying bacterial chromosome segregation are much less understood mechanistically, but are just as critical for cellular proliferation (*Badrinarayanan et al., 2015*). The starting point for bidirectional chromosomal replication, the origin (*ori*), has a crucial role in chromosome organisation and segregation. Not only is it duplicated and segregated first but its dynamic genomic position defines the position of other chromosomal regions with respect to the cell (*Duigou and Boccard, 2017*; *Junier et al., 2014*).

In new-born *Escherichia coli* cells growing under relatively slow growing conditions in which initiation of replication and its completion occur within a single cell generation, the 'home' position of the origin (henceforth and in the model, *ori*) is at mid-cell (*Nielsen et al., 2006b*; *Niki et al., 2000*; *Wang et al., 2006*). After replication, and consequent 10–15 min of 'cohesion', arising at least in part from interlinking of the two daughter chromosomes (precatenation) (*Joshi et al., 2011*; *Nicolas et al., 2014*; *Nielsen et al., 2006a*; *Nolivos et al., 2016*; *Reyes-Lamothe et al., 2008*; *Wang et al., 2008*), duplicated origins migrate rapidly to opposite quarter positions, which become the new home positions for the remainder of the cell cycle (*Cass et al., 2016*; *Kuwada et al., 2013*). Other genomic loci migrate sequentially with similar dynamics (*Cass et al., 2016*).

The mechanisms that underlie *ori* positioning and direct newly replicated sisters to opposite cell halves remain unclear (*Badrinarayanan et al., 2015*). This is particularly the case in *E. coli* and its relatives, which do not carry ParABS systems that facilitate the segregation of low copy plasmids and

some other bacterial chromosomes (*Livny et al., 2007*). However, MukBEF, a functional homolog of ubiquitous Structural Maintenance of Chromosomes (SMC) complexes (*Nolivos and Sherratt, 2014*; *Rybenkov et al., 2014*), plays a role in *E. coli* chromosome organisation-segregation. One of its functions is to recruit the type II topoisomerase Topo IV (*Li et al., 2010*; *Nicolas et al., 2014*), which is required for the timely removal of catenanes from newly replicated sister chromosomes (*Wang et al., 2008*). Under slow growth conditions, MukBEF forms a small number of dynamic clusters (visualised as fluorescent foci) located at the middle or quarter positions (*den Blaauwen et al., 2001*; *Ohsumi et al., 2001*), in close association with *ori* (*Danilova et al., 2007*), and the splitting and movement of these foci occurs concurrently with the segregation of *ori*s to the quarter positions (*Danilova et al., 2007*; *Nicolas et al., 2014*). Foci consist of on average 16 dimeric slowly-diffusing MukBEF complexes (*Badrinarayanan et al., 2012b*). The colocalisation with *ori* is not required for either MukBEF foci formation or positioning: depletion of Topo IV results in cells with multiple catenated *ori*s forming a single focus at mid-cell but with multiple MukBEF clusters positioned throughout the nucleoid (*Nicolas et al., 2014*). Thus MukBEF clusters are not necessarily assembled at or bound to *ori*, consistent with the lack of any sequence specificity (*Nolivos et al., 2016*). Furthermore, restoration of Topo IV activity leads to the decatenated *ori*s moving to the MukBEF clusters suggesting that MukBEF recruits or positions *ori* (*Nicolas et al., 2014*). Consistent with this hypothesis, depletion of functional MukBEF results in *ori* mis-positioning that is subsequently restored upon repletion (*Badrinarayanan et al., 2012a*).

If MukBEF clusters position *ori*s, what positions MukBEF clusters? Given that molecules in the clusters turnover continuously with a timescale of about one minute (*Badrinarayanan et al., 2012b*) and that MukBEF binds DNA non-specifically, how does it even form clusters? We have proposed that a self-positioning stochastic Turing pattern can explain the positioning of MukBEF clusters (*Murray and Sourjik, 2017*) (see the methods section for a review). A Turing pattern is a spatial pattern in the concentration of a reactant in a reaction-diffusion system that arises spontaneously due to a diffusion-driven instability (*Koch and Meinhardt, 1994*; *Turing, 1952*). Put simply, diffusion, rather than having a homogenising effect can actually, in combination with chemical reactions, create a spatially varying concentration profile. Such patterns are examples of self-organisation, a more general term that describes any dissipative non-equilibrium energy-dependent order that arises as a result of collective non-linear interactions (*Halley and Winkler, 2008*). We used the Turing mechanism to explain the positioning of MukBEF foci and showed that a flux-balance mechanism and stochasticity work together to ensure that a specific Turing pattern is selected: short cells consistently have a single centre-positioned peak in the MukBEF concentration, while longer cells have quarter positioned peaks.

With this model in hand, we now investigate how MukBEF clusters could position chromosomal origins. In particular, we address whether the self-organising MukBEF gradient proposed in our model has the correct properties to act as an attracting gradient for *ori*. Additionally, it is critical that each newly replicated sister *ori* is recruited to a *different* MukBEF focus, a non-trivial requirement. We find that a self-organising MukBEF gradient can indeed accurately reproduce the observed *ori* dynamics, apparent diffusion constant and drift rate. A proposed preferential loading of MukBEF within *ori* introduces a non-trivial interaction between MukBEF foci and *ori*s that leads to accurate and stable partitioning as an emergent property of the system. Importantly, the model does not contain any actual directed force. MukBEF requires energy in the form of ATP to establish a self-organised gradient but it is not pulled to the middle or quarter positions by any active force. Similarly, the attraction of *ori* up the MukBEF gradient may be due to energetic considerations and the elastic nature of the chromosome (a DNA-relay) resulting on the macro scale in an effective (rectification) force and directed motion.

## Results

### *ori* is attracted towards MukBEF foci

As discussed above, perturbative experiments support the hypothesis that MukBEF clusters position *ori*s in *E. coli* (*Badrinarayanan et al., 2012a*; *Nicolas et al., 2014*). However, it is unclear if this hypothesis is supported by the observed colocalisation of MukBEF clusters with *ori*s in unperturbed cells. It is possible that MukBEF clusters and *ori* could be positioned independently of one another

as a result of the global organisation of the chromosome with the result that they show colocalisation but without their positions being correlated. To examine this possibility, we revisited the colocalisation of MukBEF clusters and *ori*s. We used only cells with a single *ori* focus, because, unlike cells with two *ori*s, they can be grouped together without scaling by simply aligning them according to their mid-cell positions and are more amenable to statistical analysis. To enrich for such cells, we treated a strain carrying fluorescently labelled MukB and *ori* (*Figure 1a*) with DL serine hydroxamate (SHX). This structural analogue of serine triggers the stringent response thereby inhibiting DNA replication initiation (*Ferullo et al., 2009*). We then measured the position of fluorescent foci along the long axis of the cell as has been done previously (*Nicolas et al., 2014*; *Nolivos et al., 2016*) and found very similar distributions for *ori* and MukB as expected (*Figure 1b*).

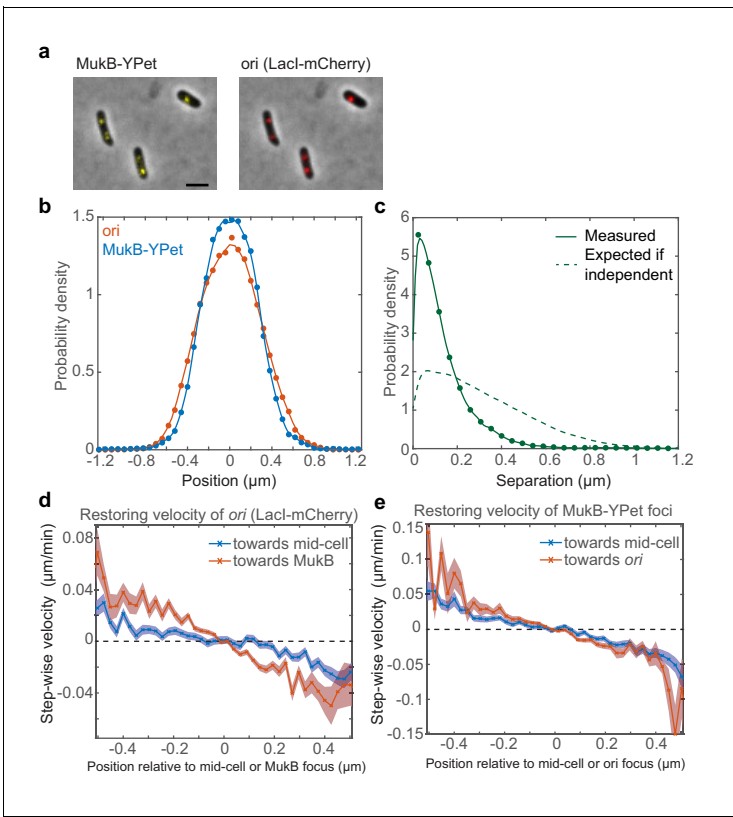

**Figure 1.** Fluorescence microscopy indicates that *ori* and MukBEF are not positioned independently of one another. A strain with FROS (LacI-mCherry) labelled *ori* and MukB-mYPet was treated with DL serine hydroxamate to obtain cells with a single non-replicating chromosome and imaged at 1 min intervals. (a) Overlay of phase contrast and fluorescence images showing three representative cells. Bar indicates 2 μm. (b) The position distribution (along the long axis of the cell) of fluorescent foci of *ori* (red) and MukB (blue). N = 31820 from 952 cells tracked over up to 56 frames. Cells have a mean length of 2.2 μm. (c) The expected distribution (dashed line) of the distance between *ori* and MukB foci given that the distributions in (b) are independent. The measured distribution (circles and solid line) of separation distances from the same cells. (d) The step-wise velocity of *ori* as a function of position relative to mid-cell (blue) and to the MukB focus (red). (e) The step-wise velocity of MukB as a function of position relative to mid-cell (blue) and the *ori* (red). Shaded regions indicate standard error. See also *Figure 1—figure supplement 1*.

DOI: https://doi.org/10.7554/eLife.46564.002

The following source data and figure supplement are available for figure 1:

**Source data 1.** Matlab .mat files containing the data used in *Figure 1*.
DOI: https://doi.org/10.7554/eLife.46564.004

**Figure supplement 1.** Positioning of *ori* (LacI-mCherry) and MukB-mYPet foci.
DOI: https://doi.org/10.7554/eLife.46564.003

To investigate if MukBEF and *ori* are positioned independently of one another, we next compared the distribution of the measured distance between them to the distribution that would be expected *if* they were positioned independently (the null hypothesis). This latter distribution is obtained by randomly selecting pairs of positions from the two measured position distributions and calculating the distance between them. When we did this, we found that MukBEF foci and *ori* are much more colocalised than would be expected if they were positioned independently (*Figure 1c*). This was confirmed by the relatively strong positive correlation (r = 0.8) between MukBEF and *ori* positions (using a robust correlation based on the median absolute deviation [*Shevlyakov, 1997*]). Importantly, the result was not due to treatment with SHX (*Figure 1—figure supplement 1b*).

If *ori* is indeed positioned by MukBEF, then we should be able to detect this in wild-type cells. In particular, we can measure the step-wise velocity of *ori* as a function of its position along the long axis of the cell. This 'restoring' velocity characterises the restoring force pulling *ori* back towards mid-cell. We can similarly determine the restoring velocity of *ori* towards MukBEF foci by measuring the step-wise velocity of *ori* as a function of position relative to the MukBEF focus. Comparing these two profiles, we found that *ori* experiences a greater restoring velocity towards the MukBEF focus than towards mid-cell (*Figure 1d*). This indicates that *ori* is not attracted to mid-cell per se, rather it is more likely attracted to the MukBEF focus, which happens to be positioned at mid-cell. Hence, together with previous results, these data strongly indicate that MukBEF positions *ori* in *E. coli*.

We next asked whether the relationship is bi-directional i.e. is MukBEF positioning affected by *ori* positioning? When we examined the restoring velocity of MukBEF foci, we found that they displayed similar biases towards *ori* and mid-cell (*Figure 1e*), suggesting that MukBEF foci are equally attracted to mid-cell and *ori* and that therefore the attraction between MukBEF and *ori* may be indeed be bi-directional. We will return to this result later.

The above results also confirm that *ori* has a special relationship with MukBEF compared to other genetic loci. This is supported by the observation that co-localisation with MukBEF is strongest for *ori* and becomes progressively weaker for *ori*-distant loci (*Danilova et al., 2007*). What is the nature of this relationship? This has been an open question for many years, despite the application of tools such as chromatin immunoprecipitation (*Nolivos et al., 2016*) and single-molecule microscopy (*Badrinarayanan et al., 2012b*; *Zawadzki et al., 2015*). In the following, we take an abductive reasoning approach common in theoretical physics. We make a starting assumption or ansatz for the nature of the MukBEF-*ori* relationship in order to build a computational model of the system and take confirmation of model predictions as evidence in support of this ansatz. In particular, we propose that MukBEF is preferentially loaded onto the chromosome within the *ori* region. This is the case for SMC, a distant relative of MukBEF (see *Gruber, 2018* for a review). In *Bacillus subtilis* and other bacteria SMC is loaded onto the chromosome at *parS* sites by the protein ParB. While no analogue of ParB/*parS* has yet been discovered in *E. coli*, in the following we will focus on exploring the effect of preferential loading and examining whether it gives results that are consistent with experimental observations. We discuss other plausible scenarios in the discussion.

## Model of *ori* positioning by self-organised MukBEF reproduces mid-cell positioning

As a first step in building a model of *ori* positioning, we incorporated the *ori* into our previous stochastic model of MukBEF self-organisation and positioning (*Murray and Sourjik, 2017*), reviewed in the methods section and illustrated in *Figure 2a*. Briefly, MukBEF exists in three states corresponding to different conformations and associations with DNA, a well-mixed cytosolic fraction and two DNA-associated states. The differing diffusion constants and nonlinear interaction between the two latter states leads to the spontaneous formation of dynamic MukBEF foci (to be understood as regions of high density) via the Turing mechanism. The positions of these foci are determined by the balancing of fluxes originating from the well-mixed cytosolic state (*Figure 2b*). The flux of molecules (number per second) reaching the MukBEF focus is proportional to the length of the nucleoid on each side since the flux of molecules arriving from the cytosol is proportional to these lengths. Thus, if the MukBEF focus is off-centre (in the case of a single focus), it experiences a differential in the incoming fluxes from either side, resulting in movement toward the equilibrium position (the centre). This flux-balance mechanism was first described in the context of plasmid positioning (*Ietswaart et al., 2014*; *Murray and Howard, 2019*; *Sugawara and Kaneko, 2011*) but is valid quite generally. Note that in this model, the chromosome is not modelled explicitly. While one could

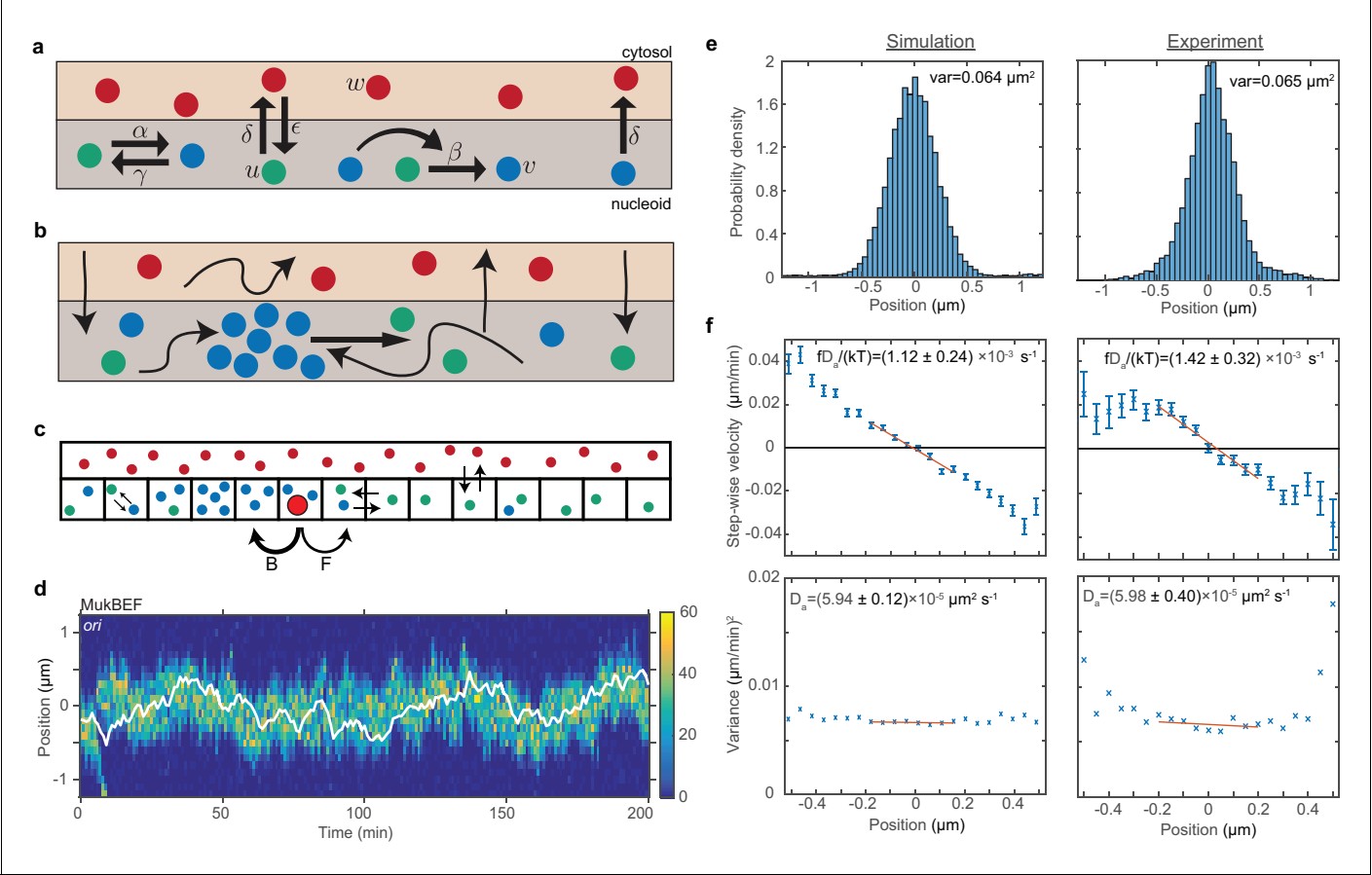

**Figure 2.** *ori* positioning by a self-organised protein gradient reproduces experimental results. (**a**) Schematic showing the reactions of the previous MukBEF model (***Murray and Sourjik, 2017***) (see methods). Species *w* diffuses in the cytosol (red). Species *u* (green) and *v* (blue) diffuse on the nucleoid. Binding and species interaction are indicated by arrows. Diffusion is not shown. See the methods for a review of the model and the model parameters. (**b**) Schematic showing the flux-balance mechanism. The thinner arrows represent binding/unbinding and diffusion. Species *w* (red) is well-mixed and therefore converts to species *u* (green) uniformly across the nucleoid. If a molecule of species *u* explores a sufficiently large region of the nucleoid before it detaches again, then the flux of *u* molecules reaching a high density region (focus) of species *v* (blue) from either side is proportional to the length of the nucleoid on either side. This difference in fluxes leads to net movement of the self-organised focus towards the position at which the fluxes balance, the mid-cell in the case of a single focus. (**c**) The stochastic model is implemented using the spatial Gillespie method which discretises the spatial dimension into compartments in which molecules react and between which molecules can diffuse. Colours label species as in (**a**). The cytosolic species is taken to be well-mixed and its concentration is therefore not simulated spatially. This is the same implementation as was used previously (***Murray and Sourjik, 2017***). In this work, we extend these simulations by incorporating the *ori* as a single diffusing particle (outlined red circle). However, unlike MukBEF its diffusion is biased, being determined by forward (**F**) and backward (**B**) jump rates that depend on the gradient of MukBEF concentration (blue circles, *v*) (see methods). (**d**) Kymograph from a single simulation showing the number of MukBEF molecules (colour scale) and the position of the *ori* (white line). (**e**) and (**f**) A comparison between the experimental data of ***Kuwada et al. (2013)*** and the results of simulations in the case of a single *ori* and 6x preferential loading. (**e**) Histograms of *ori* position (unscaled) along the long axis of the cell. Zero is the middle position. (**f**) Mean (top) and variance (bottom) of the step-wise velocity as a function of position relative to mid-cell. Bars indicate standard error. The linear velocity profile at mid-cell is indicative of diffusion in a harmonic potential ($V(x) = \frac{1}{2}fx^2$). In such a model the variance of the step-wise velocity is independent of position. Thus we obtain the apparent diffusion constant $D_a$ and drift rate $d_a = \frac{fD_a}{kT}$ by fitting to the central region. Bounds are 95% confidence intervals. Red lines are weighted linear fits. Simulated data are from 100 independent runs, each of 600 min duration. Experimental data are based on over 16000 data points from 377 cells. Both data sets use 1 min time-intervals. Simulations are from 2.5 μm cells, whereas experimental data is from a range of cell lengths. See methods for further details and model parameters. See also ***Figure 2—figure supplement 1–5***.

DOI: https://doi.org/10.7554/eLife.46564.005

The following figure supplements are available for figure 2:

**Figure supplement 1.** Compartment size effects.
DOI: https://doi.org/10.7554/eLife.46564.007

**Figure supplement 2.** Properties of MukBEF model and *ori* positioning.
DOI: https://doi.org/10.7554/eLife.46564.006

*Figure 2 continued on next page*

*Figure 2 continued*

**Figure supplement 3.** Fitting the model to experimental data.
DOI: https://doi.org/10.7554/eLife.46564.008
**Figure supplement 4.** Additional properties in short cells with one *ori*.
DOI: https://doi.org/10.7554/eLife.46564.009
**Figure supplement 5.** MukBEF is attracted towards mid-cell at low loading ratios and towards *ori* at high loading ratios.
DOI: https://doi.org/10.7554/eLife.46564.010

theoretically use a combined particle and polymer based approach, such simulations are not yet feasible. Rather, the action of condensins is typically implemented in polymer simulations implicitly (*Fudenberg et al., 2016*; *Miermans and Broedersz, 2018*). However, we are explicitly interested in the fact that MukBEF forms discrete positioned foci. We therefore take a protein-centric approach and model the chromosome implicitly but MukBEF explicitly.

We treat the *ori* as a diffusing particle, the movement of which is biased in the direction of increasing MukBEF concentration (*Figure 2c*) (see methods for details). That is, the probability is higher that *ori* will move up the MukBEF gradient than down it. We perform the simulations in one dimension, representing the long axis of the cell, the dimension along which positioning and segregation occur. This is justified as both *ori* and MukB are confined within the transverse direction to the centre region of the cell: 95% of foci are within the centre 40% (300 nm) of the cell width (*Figure 1—figure supplement 1d–g*). MukBEF is also a very large molecule complex, with the arc length from its hinge domain to either of its two head domains being about 70 nm (*Matoba et al., 2005*). This suggests it can be treated as operating on a coarser level than individual strands of DNA. However the primary reason is due to an inherent limitation of the Reaction Diffusion Master Equation method for non-linear reactions. Every voxel (compartment) must be well-mixed and this condition is violated by non-linear reactions at small voxel volumes (*Isaacson, 2009*; *Isaacson and Peskin, 2006*). While this can be overcome for bimolecular reactions (*Erban and Chapman, 2009*; *Hellander and Petzold, 2016*; *Isaacson, 2013*), there is currently no such remedy for tri-molecular reactions, as present in our (and most) Turing models. As a quantification of the resulting artefacts, we measured the mean total number of species *v* as a function of compartment size (*Figure 2—figure supplement 1*). While the mean number is stable in one dimension, it decreases rapidly in two and three dimensions as the compartment size is decreased. One might hope for a range of compartment sizes, small enough to realise the geometry but large enough to avoid small compartment-size effects. However, no such range exists. We will therefore confine our stochastic simulations to one dimension. However, as we shall show, this limitation does not prevent us from explaining the observed experimental behaviour and making falsifiable predictions.

We take initially the case of short cells (2.5 μm) with a single *ori*. For the moment, we do not implement the effect of preferential loading at *ori* on MukBEF dynamics. We found, as expected, that *ori* tracks the self-organised MukBEF focus, resulting in mid-cell positioning (*Figure 2d* and *Figure 2—figure supplement 2b*). To more carefully examine the directed movement of *ori*, we measured the *ori* velocity as a function of position as we did previously (*Figure 1d*). Given the self-organising and fluctuating nature of the MukBEF gradient in our model (which is representative of the in vivo behaviour), it was not obvious that the model would reproduce the observed relationship. However, we indeed found a similarly linear velocity profile. Furthermore, quantitative comparison of our simulations with the experimental data from *Kuwada et al. (2013)* was carried out by fitting the data in the mid-cell region to a theoretical model of diffusion in a harmonic potential, thereby obtaining an apparent diffusion constant and drift rate. We found that by adjusting only two parameters, the *ori* diffusion constant and drift parameter, we were able to obtain good agreement with the values obtained from the experimental data (*Figure 2—figure supplement 3b*). However, the resultant position distribution did exhibit somewhat fatter tails (*Figure 2—figure supplement 3a*).

This fitting of the experimental data also allows us to estimate the spring-like force on the *ori*. At a distance $x$ from mid-cell, the force is given by $F = -fx$, where $f$ is obtained from the slope and variance of the velocity profile (*Figure 2—figure supplement 3b*). At 0.2 μm from mid-cell this gives a restoring force of 0.02 pN, similar to the value measured in an in vitro reconstitution of a plasmid partitioning ParABS system (*Vecchiarelli et al., 2014*). Note that the data in Kuwada et al. are from

growing cells and we use them rather than our data in *Figure 1* for consistency with later simulations that incorporate growth.

In the previous simulations *ori* moves up the MukBEF gradient but has no effect on the MukBEF gradient itself as we have not yet implemented that MukBEF is preferentially loaded onto the DNA within the *ori* region. In previous work, we showed that preferential loading at a fixed spatial location perturbs the positioning of the self-organised MukBEF foci due to the modified flux differential across foci (*Murray and Sourjik, 2017*). In the case of a single MukBEF focus, the equilibrium position is no longer at mid-cell but somewhere between mid-cell and the location of preferential loading, depending on the strength of the loading. Thus the presence of a preferential loading site in the *ori* should lead to an effective mutual attraction between *ori* and MukBEF foci i.e. *ori* is attracted up the MukBEF gradient, while at the same time the' home' position of the MukBEF focus is shifted towards *ori*. We expected this to increase the association between the two and reinforce mid-cell positioning.

We added preferential loading into the simulations by increasing the loading rate of MukBEF in the compartment containing the *ori* relative to the other compartments while keeping the overall loading rate unchanged. This was observed to have a suppressive effect on noise (*Figure 2—figure supplement 4a*). At intermediate levels of preferential loading, the positions of both MukBEF and *ori* deviate less from mid-cell. Looking at individual simulations we could see that *ori* rarely escapes the MukBEF focus, rather the focus tracks *ori* and brings it back to the middle position. As a result *ori* only rarely undergoes diffusive excursions away from MukBEF and its home position as were observed without preferential loading. The reduction in the variance of the position distributions was reversed at higher loading ratios (*Figure 2—figure supplement 4a*). We also found that preferential loading resulted in stronger colocalisation of *ori* with the MukBEF focus and this led to excellent agreement with previous measurements of their separation distance (*Figure 2—figure supplement 4b*). Note that this experimental data was not used to constrain the model and this agreement thus constitutes confirmation of a model prediction and support for preferential loading.

We next examined if the simulations could reproduce the observed experimental velocity profiles (*Figure 1d,e*). We found that *ori* shows a stronger restoring velocity towards MukBEF than mid-cell and this was independent of the level of preferential loading (*Figure 2—figure supplement 5d–f*). The restoring velocity to mid-cell is non-zero due to the fact that MukBEF fluctuates around mid-cell. These results were to be expected as the biased motion of *ori* up the MukBEF gradient is explicitly included in the simulations. What was less clear was how the MukBEF focus would behave. We found that without preferential loading MukBEF displays a stronger restoring velocity towards mid-cell (*Figure 2—figure supplement 5g*) consistent with its positioning by the flux-balance mechanism. At high loading however, it shows a stronger bias towards *ori* (*Figure 2—figure supplement 5i*) consistent with the previous observation that the attraction to *ori* dominates, while at intermediate levels (6x) we found a very similar restoring velocity to both *ori* and mid-cell as was observed in the corresponding experimental profiles (*Figure 1e*). This is also the same level of preferential loading that leads to the most robust positioning (*Figure 2—figure supplement 4a*). Thus, an intermediate level of preferential loading appears to be most consistent with experimental observations.

Given that preferential loading was found to have an effect on the apparent diffusion constant and drift rate (*Figure 2—figure supplement 4c*), we needed to refit the model to the experimental *ori* velocity data. To do so we chose a particular value for preferential loading ratio, the one that minimised the variance, 6x (this choice will be justified in the section). We were able to find new values for the diffusion and drift parameters that lead to excellent agreement with the experimental values (*Figure 2f*). Furthermore, the resulting distribution of the *ori* positions showed better agreement with the experimental distribution, with the fat tails observed in the absence of preferential loading no longer present (*Figure 2e*).

## Preferential loading leads to stable and accurate partitioning

While promising, the above results are not sufficient to suggest that MukBEF can explain the in vivo behaviour of *ori*. The challenge arises after *ori* has been replicated. A true partitioning mechanism must ensure that each replicated *ori* is maintained at a *different* quarter position. A simple gradient based mechanism cannot, a priori, satisfy this requirement as both *ori*s could just as easily move towards the same quarter position. Furthermore, the experimental data suggests that once *ori*s separate they do not subsequently interchange their positions (cross paths). This ordering is essential

during multi-fork replication, where the multiple *ori*s of each segregated chromosome must be positioned to the appropriate cell half to avoid guillotining the chromosome upon cell division.To examine if the model is capable of accurate and ordered partitioning, we performed simulations with two *ori*s in longer cells of 5 μm, in which MukBEF self-organises into, on average, two foci, one at each quarter position. With or without preferential loading, the average profile of *ori* positions displayed two peaks centred on the quarter positions (*Figure 3d*, blue line). However, we found that without preferential loading approximately half of the individual simulations have both *ori*s near the same quarter position (*Figure 3a*), clearly indicating that partitioning was not accurate. This was the case even though the simulations were initialised with *ori*s at opposite quarter positions. Evidently, a model of *ori* simply moving up the MukBEF gradient is not sufficient to explain partitioning as the noise inherent to the system means that it can switch stochastically between partitioned and un-partitioned states.

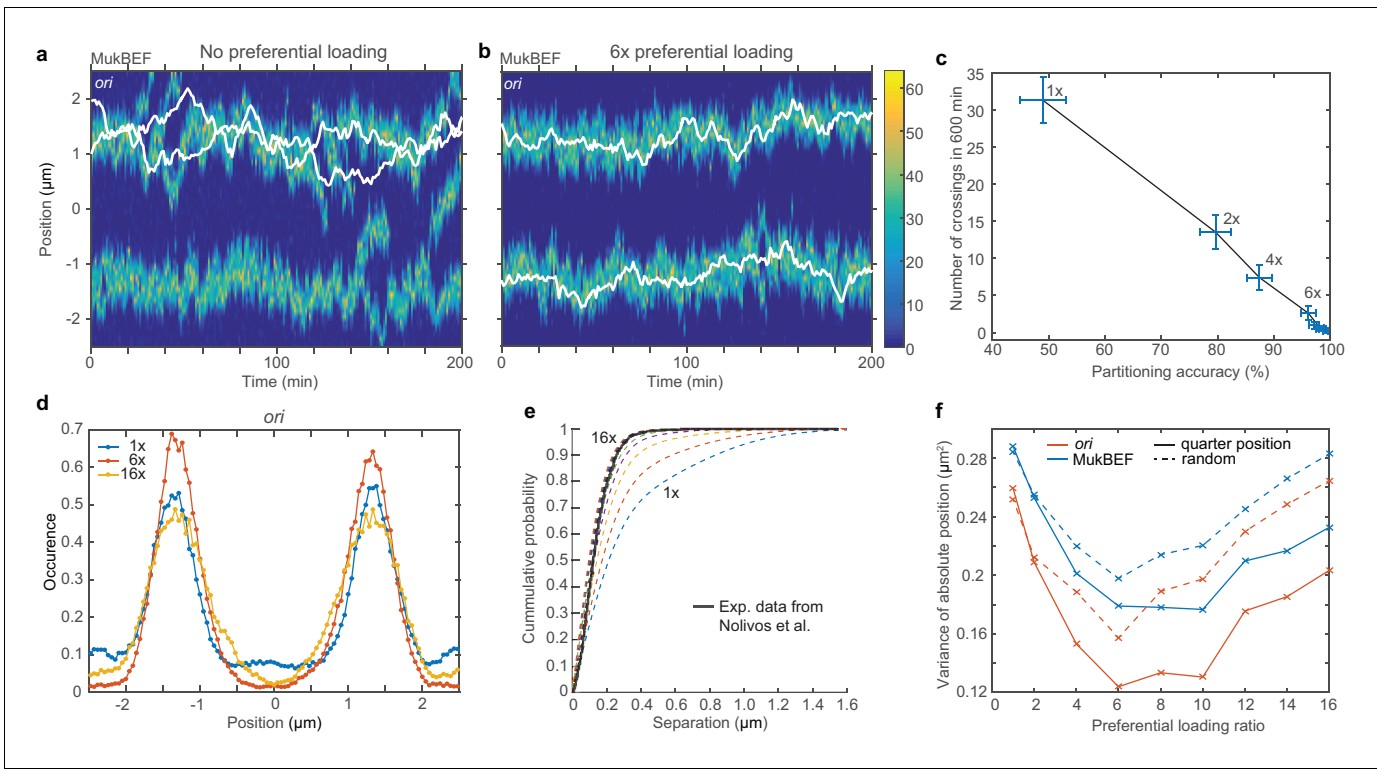

**Figure 3.** Preferential loading of MukBEF at *ori* leads to correct and stable partitioning. (a) Example simulated kymograph showing two *ori* (white lines) diffusing around the same MukBEF peak (colour scale). This occurs approximately 50% of the time. (b) The addition of 6x preferential loading of MukBEF at *ori* positions results in correct partitioning of *ori*. The loading rate in each of the spatial compartments containing *ori* is six times that of the other 48 compartments. The total loading rate is unchanged. (c) Partitioning accuracy is measured by the fraction of simulations with *ori* in different cell halves. Stability is measured by the number of times *ori* cross paths. Both partitioning accuracy and stability increase with preferential loading up to approximately 6x. Preferential loading ratios are as in (f). Points and bars indicate mean and standard error over independent simulations. See also *Figure 3—figure supplement 1c*. (d) Histograms of *ori* positions for 1x, 6x and 16x preferential loading. Positioning is more precise at 6x than with no or 16x preferential loading. See *Figure 3—figure supplement 1a* for MukBEF distributions. (e) The cumulative probability distribution for the separation distance between *ori* and MukBEF peaks. Experimental data (black line) is from *Nolivos et al. (2016)*. The addition of preferential loading leads to substantially better agreement. Preferential loading ratios are as in (f). See also *Figure 2—figure supplement 4b*. (f) The variances of individual peaks (obtained by reflecting the data around the mid-position) have a minimum at approximately 6x preferential loading. Solid lines are from simulations with *ori* initially at the quarter positions, as for (a)-(e). Dashed lines are from simulations with random initial *ori* positions. Simulations were performed for a 5 μm domain and two *ori*.

DOI: https://doi.org/10.7554/eLife.46564.011

The following figure supplement is available for figure 3:

**Figure supplement 1.** Additional properties of model in longer cells with two *ori*.

DOI: https://doi.org/10.7554/eLife.46564.012

However we found that preferential loading resulted in stable and accurate partitioning (*Figure 3b*). A preferential loading ratio greater than six (i.e. six times more loading than elsewhere) was sufficient to ensure that one and only one *ori* was positioned to each quarter position and they do not interchange (*Figure 3c*). These simulations had *ori*s initialized at opposite quarter positions (so as to investigate the intrinsic stability of that configuration). However, the bias of the system towards the desirable quarter-positioned configuration was present when both *ori*s were initialized at mid-cell or at random positions (*Figure 3—figure supplement 1c,d*). While configurations with both *ori*s associated to the same MukBEF peak occurred more frequently under these conditions, in the presence of sufficient preferential loading the system eventually and irreversibly transitions to the quarter positioned configuration.

As previously observed in simulations of short cells, we found that preferential loading results in stronger colocalisation of *ori*s with MukBEF foci (*Figure 3e*) and has a suppressive effect on noise at intermediate ratios with MukBEF foci and *ori* deviating less from the quarter positions (*Figure 3f*). However, partitioning accuracy remained robust even at high preferential loading ratios (*Figure 3c*). Looking at individual simulations, we observed that the nature of the variance was different. While the number of foci is maintained accurately at two (*Figure 3—figure supplement 1b*) and the foci are tightly associated to each *ori* (*Figure 3e*) they are together more mobile than at intermediate ratios. Effectively, the MukBEF clusters begin to follow *ori*, rather than the other way around.

These results demonstrate that preferential loading of MukBEF changes the stability of the different steady states of the system. In its absence, the desirable (*ori* associated to opposite quarter-positioned MukBEF peaks) and undesirable (both *ori*s associated to the same MukBEF peak) configurations have equal likelihood, as measured by partitioning accuracy (*Figure 3c*, *Figure 3—figure supplement 1d*), and the system can stochastically jump from one state to the other. As preferential loading is increased, the desirable configuration becomes more stable until the system is found almost exclusively in that state.

## Accurate partitioning during growth

The previous simulations were of non-growing cells and of long duration. While, they were useful to examine the intrinsic stability of the different states in order to understand why *ori*s *remain* partitioned, they do not demonstrate that our model can explain how *ori*s *become* partitioned within the timescale and setting of a growing cell. We therefore incorporated exponential growth and *ori* replication into our simulations. The former was implemented by randomly adding a new spatial compartment after every time interval corresponding to growth by one compartment length (0.1 μm). The *ori* was duplicated at a randomly chosen time-point obtained from an experimentally derived distribution (the mean time of duplication was 40 min into the cell cycle). After duplication, the compartment that previously contained a single *ori,* then contains two *ori*s, which are free to move independently of each other (but dependent on the local MukBEF concentration).

We first examined growth in the absence of preferential loading. Similarly to what we observed previously in the simulations of non-growing cells (*Figure 3a*), we found that duplicated *ori*s often remained associated to the same MukBEF focus, resulting in a partitioning accuracy (defined as before as the fraction of simulations with *ori*s in opposite cell halves) of only 25% by the end of the cell cycle (120 min) (*Figure 4—figure supplement 1a,b*). When we introduced preferential loading at *ori*, we found firstly that it delayed the splitting of MukBEF foci, similar to a spatially fixed loading site (*Figure 4—figure supplement 1d*) (*Murray and Sourjik, 2017*). The feedback from *ori* to MukBEF, nonetheless resulted in somewhat improved (39%) partitioning. The effect was similar to what we observed in the simulations of non-growing cells. Preferential loading promotes partitioning (and colocalisation) but sufficient time is required for the system to stochastically jump out of the undesirable configuration. But once it does the quarter positioned configuration is stable and does not revert back. The long runtime of the previous simulations meant the system had sufficient time to transition (*Figure 3—figure supplement 1d*) but this is not the case here.

Clearly, this level of partitioning accuracy is not representative of the biological situation, where ~ 95% of cells have partitioned *ori* already 20 min after initial separation, with *ori*s being separated at that point by an average distance of 33% of the cell length. We wondered whether the poor partitioning observed in the simulations could be overcome by the introduction of polymeric effects. Entropic repulsion is believed to play a role in chromosome segregation and organisation (*Jun and Mulder, 2006*; *Jun and Wright, 2010*; *Jung et al., 2012*; *Minina and Arnold, 2014*;

*Youngren et al., 2014*). In particular, newly replicated *ori*s would experience the entropic repulsion of two closed loops. This is consistent with experimental observations, in which duplicated *ori*s (and indeed all loci) experience an initially large segregation velocity (*Cass et al., 2016*; *Lampo et al., 2015*). Numerical studies have demonstrated that the effective potential associated with such a repulsion has the approximate form of a (half) Gaussian in the centre-of-mass separation (*Bohn and Heermann, 2011*; *Bohn and Heermann, 2010a*). We therefore incorporated entropic repulsion into the simulations by adding a repulsive force between *ori*s specified by such a potential. Important to note is that this force is short-range and therefore, with a small enough value for the range, it does not affect the desirable quarter-positioned configuration but rather acts to destabilise the undesirable configuration having both *ori*s associated to the same MukBEF focus.

This introduced two unknown parameters, the depth of the potential and its range. We performed a sweep over these parameters and measured the partitioning accuracy 20 min after *ori* duplication (*Figure 4—figure supplement 2*). We found that entropic repulsion on its own (i.e. no preferential loading) was not able to reproduce the observed behaviour. Increasing the range of the repulsion to 400 nm, which is likely unphysical, did lead to accurate partitioning but at the cost of *ori*s that were too far separated (*Figure 4—figure supplement 2*), especially immediately after duplication. Furthermore, as we observed previously, without preferential loading the *ori* can escape from the MukBEF peaks resulting in weaker colocalisation, as well as there being significant noise in the both the number and position of MukBEF peaks. Repulsion does not change these effects (*Figure 4—figure supplement 3*). As already noted, preferential loading on its own was also insufficient for accurate partitioning (*Figure 4—figure supplement 2*).

On the other hand, combining preferential loading with short-range entropic repulsion of *ori*, gave the properties of both and allowed the model output to move much closer to the measured partitioning accuracy and relative separation (*Figure 4—figure supplement 2*) and resulted in kymographs with the experimentally observed behaviours (*Figure 4a–d*). The short-range entropic repulsion destabilises the undesirable configuration so that system switches, in a timely manner, to the quarter positioned configuration. As we saw for non-growing cells, preferential loading stabilises this configuration, keeping *ori* in close association with their corresponding quarter-positioned MukBEF peak, thereby preventing both diffusive excursions and any attempts to return to the undesirable configuration.

While the output looks qualitatively promising, we wanted to make a quantitative comparison of *ori* dynamics. Therefore, we compared the simulated time-courses (using the preferential loading and the range and strength of *ori* repulsion suggested from the parameter sweep) with previous experimental results. To reduce the dimension of the data and accommodate the variation in cell length and cell cycle duration, we examined the dynamics of *ori* from the time of duplication (simulated data) or the time of initial *ori* foci separation (experimental data [*Kuwada et al., 2013*]). We found good agreement between two cell length independent measures: the segregation velocity (the change of the absolute distance between *ori*s between time points) and the partitioning accuracy (*Figure 4e,f*). Thus, with the addition of entropic repulsion, the model is capable of reproducing the observed *ori* dynamics in growing cells.

## Directed movement of *ori* can arise from spatially-dependent looping interactions

Our experimental data indicates, and our model assumes, that the *ori* experiences directed movement up the gradient constituting a MukBEF focus. How could such an attraction arise? It has previously been argued in the DNA relay and Brownian ratchet models of partition complex positioning by the ParABS system (*Hu et al., 2017*; *Hu et al., 2015*; *Lim et al., 2014*; *Surovtsev et al., 2016*) that the elastic nature of the chromosome itself (*Wiggins et al., 2010*) can be harnessed to power directed motion of partitioning complexes. The elastic fluctuations of the chromosome allow partitioning complexes to detect local differences in ParA-ATP, a protein that tethers them non-specifically to the nucleoid. The result is that complexes move in the direction of greatest ParA-ATP concentration. However, this idea has never been tested polymerically. This is critical for migrating *ori*, since, unlike plasmids, the *ori* would experience an entropic counter force due to the polymeric nature of the chromosome. Nevertheless, we wondered whether a similar mechanism might underlie the biased movement of *ori* towards MukBEF foci.

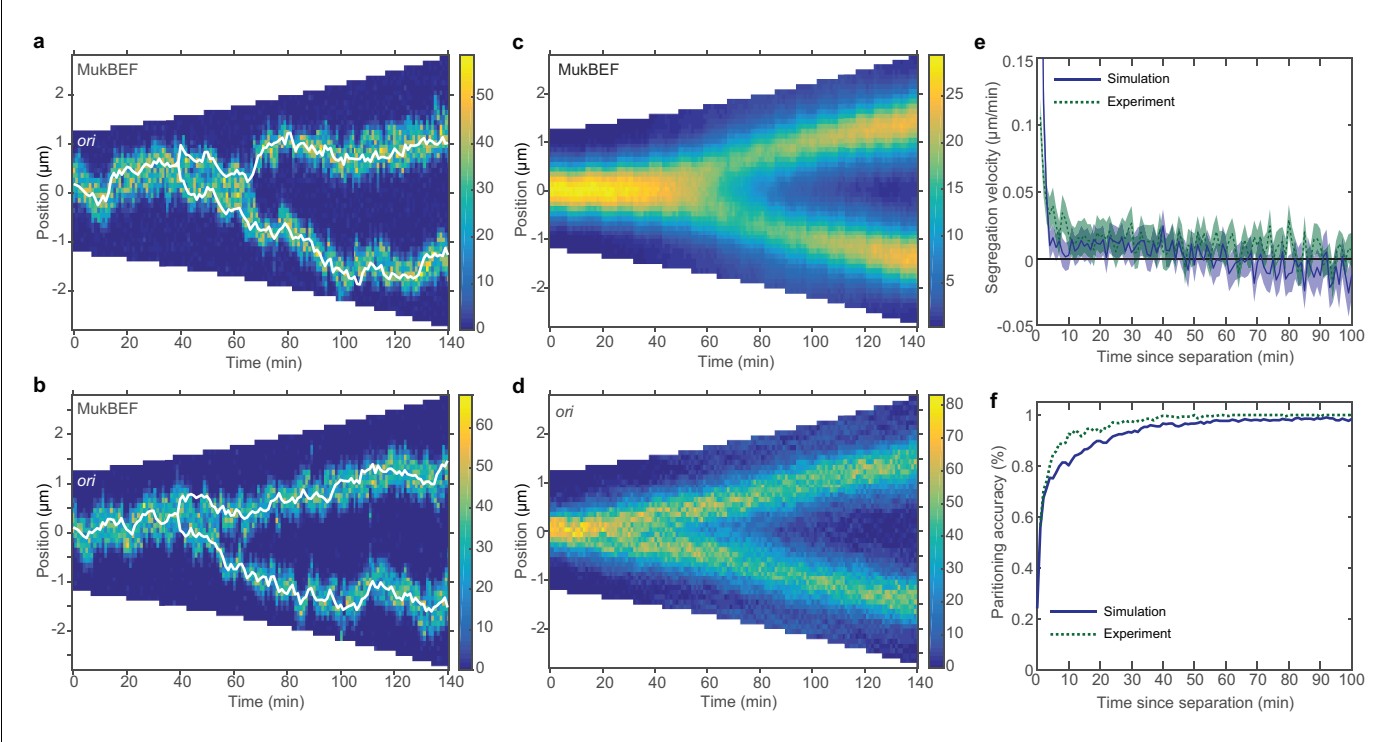

**Figure 4.** Repulsion between newly replicated *ori* results in realistic simulations of growing cells. (**a**) and (**b**) Two example kymographs from individual simulations during exponential growth (doubling time of 120 min) in the presence of a repulsive force between *ori*. Shown is the number of MukBEF molecules (colour scale) overlayed with the *ori* position (while lines). (**c**) and (**d**) Average kymographs of MukBEF (**c**) and *ori* position (**d**). (**e**) and (**f**) Segregation velocity (the step-wise rate of change of the absolute distance between *ori*) (**e**) and partitioning accuracy (**f**) plotted as function of the time since *ori* duplication (simulations, blue) or separation (experiment, green). Experimental data is from *Kuwada et al. (2013)*. Shading indicates 95% confidence intervals. The segregation velocity has been corrected for growth. Simulation results in (**c–f**) are from 450 independent simulations and use 10x preferential loading ratio and a repulsion range of 200 nm (as indicated by the black circle in *Figure 4—figure supplement 2*).
DOI: https://doi.org/10.7554/eLife.46564.013

The following figure supplements are available for figure 4:

**Figure supplement 1.** The effect of preferential loading during growth.
DOI: https://doi.org/10.7554/eLife.46564.014

**Figure supplement 2.** Preferential loading and entropic repulsion are individually not capable of reproducing the observed partitioning efficiency.
DOI: https://doi.org/10.7554/eLife.46564.015

**Figure supplement 3.** Repulsion between *ori* alone results in incorrect dynamics.
DOI: https://doi.org/10.7554/eLife.46564.016

In particular, we wondered whether directed movement of *ori* can arise due to the DNA bridging activity of MukBEF (*Petrushenko et al., 2010*). It has recently been demonstrated in vivo that Muk-BEF promotes long-range DNA interactions (*Lioy et al., 2018*). Given the association between Muk-BEF and *ori*, it is plausible that MukBEF preferentially forms DNA contacts involving the *ori* region. As such contacts would reduce the mobility of the DNA polymer, we would expect that *ori* would colocalise with MukBEF foci. To study this possibility, we turned to polymer simulations. We modelled the chromosome as a self-avoiding ring polymer confined in a rectangular cuboid and used the dynamic loop model (*Bohn and Heermann, 2010b*) to mimic the formation of DNA loops (bridges) between *ori* (a specific monomer of polymer chain) and distant DNA sites (other monomers) (see methods for details). As it is not computationally feasible to explicitly include the reaction-diffusion dynamics of MukBEF into the polymer simulations, we instead incorporated MukBEF implicitly via a spatially dependent looping probability along the long axis of the cuboid (nucleoid) representing the MukBEF concentration profile (*Figure 5a*). We found that this resulted in the *ori* being positioned to the middle of the nucleoid, where the looping probability was greatest (*Figure 5b*, blue line). This was in contrast to the uniform position distribution observed when a uniform looping

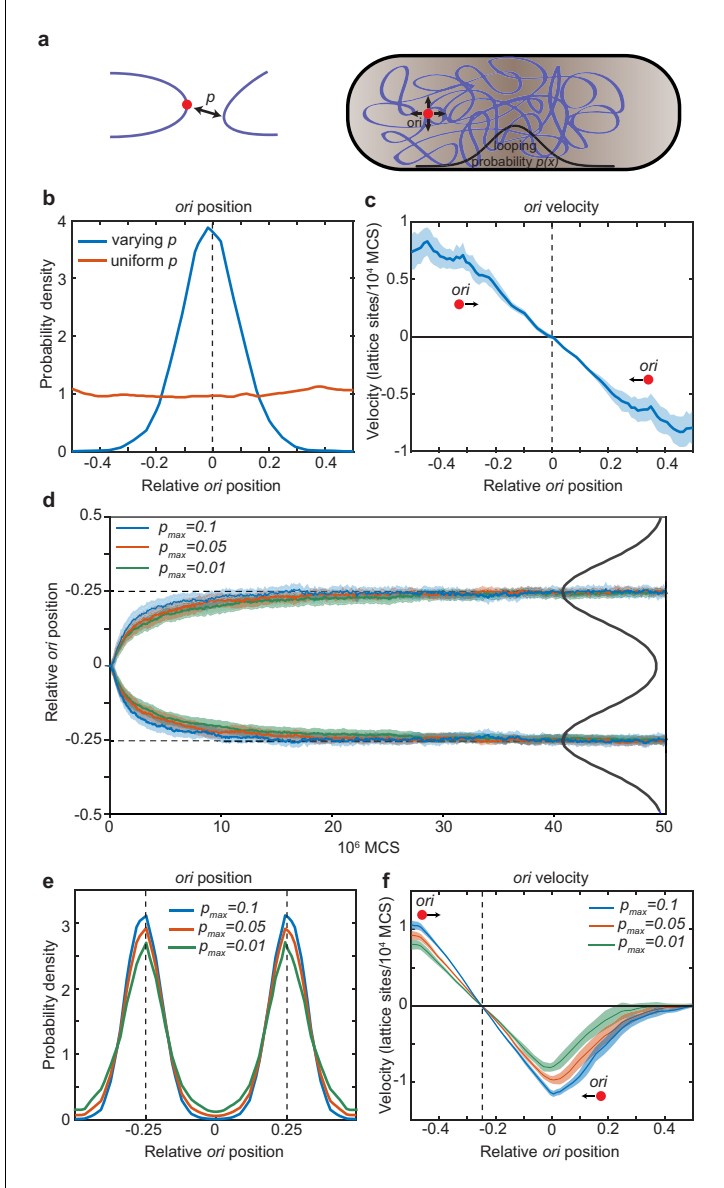

**Figure 5.** Directed movement of *ori* can arise from spatially-dependent looping interactions. (**a**) A diagram illustrating how the elastic fluctuations of *ori* allow it sample the spatial looping probability distribution. It is therefore more likely to form a loop with a locus that is closer to mid-cell, where the probability of looping, *p*, is higher. (**b**) Probability density of relative *ori* position along the long axis of the cuboid (aspect ratio 4:1) with (blue) and without (red) a spatially-varying looping probability (a Gaussian shaped distribution centred at 0 with standard deviation 0.1 in units of long-axis length; the looping probabilty at 0, $p_{max}$, is 0.02.). See also *Figure 5—figure supplement 1*. (**c**) The mean *ori* velocity along the long axis as a function of relative position. Shaded area indicates standard error. In (**b**) and (**c**), the *ori* position was read out every 50000 Monte Carlo time-steps (MCS) and data is from 50 independent simulations with approximately 10000 data points from each. (**d**) The mean relative position of *ori*s along the long axis of the cuboid during segregation. Simulations were initialized with two overlapping polymers with the *ori* monomers at the middle position. We used a looping probability distribution (black line) with the shape of the sum of two Gaussians centred at the quarter positions with standard deviation 0.1 in units of long-axis length. Results for different values of the looping probability at the quarter positions, $p_{max}$, are shown. Data is from 500 independent simulations read out as in (**c**). Shading indicates the standard error. (**e**) Probability density of relative *ori* positions in simulations of two polymers described in (**d**) after equilibration i.e. the polymers have segregated to opposite ends of the cuboid. (**f**) The mean step-wise *ori* velocity for one of the two segregated polymers. This polymer is confined to the left side of the cuboid. The *ori* experiences a restoring velocity to the approximate −1/4 position. The right half of the curve is due to infrequent excursions of the *ori* into

*Figure 5 continued*

the other half of the cuboid. The shaded region indicates standard error. See also *Figure 5—figure supplement 2*.

DOI: https://doi.org/10.7554/eLife.46564.017

The following figure supplements are available for figure 5:

**Figure supplement 1.** Positions of other loci.

DOI: https://doi.org/10.7554/eLife.46564.018

**Figure supplement 2.** The mean *ori* velocity for the polymer not shown in *Figure 5f*.

DOI: https://doi.org/10.7554/eLife.46564.019

probability was applied (red line). We also found that the positioning of *ori* affected the organisation of the entire polymer, which took with up a left-*ori*/*ter*-right configuration (*Figure 5—figure supplement 1*), consistent with previous results on the effect of a forced localisation of *ori* (*Junier et al., 2014*).

We next asked whether the distribution of the *ori* arises as a time-average or whether the movement of *ori* is directed. When we examined the velocity of the *ori* as a function of the long-axis position, we found that the *ori* indeed experiences a restoring velocity towards mid-cell i.e. directed movement (*Figure 5c*). We envisage this working as follows. On a short timescale, the *ori* fluctuates about its current 'home' position. This allows it to locally sample the spatially-varying looping probability. It is then most likely to form a loop with another monomer in the direction in which the looping probability is greatest i.e. the direction of greatest MukBEF. The polymer subsequently relaxes, the *ori* is released to a new 'home' position and the cycle repeats. In this way, elastic fluctuations of the polymer power the movement of *ori* up the gradient in the looping probability. Thus, directed movement of *ori* up the MukBEF gradient can plausibly arise due to a MukBEF-mediated, spatially-varying looping probability.

Finally, we examined how spatially-dependent looping affects chromosome segregation and the quarter positioning of duplicated *ori*s. We simulated two chromosomes, initially overlapping with their *ori*s at mid-cell, in the presence of a bi-modal looping probability distribution (with a peak at each quarter position). In the absence of looping, entropic repulsion ensures that the *ori*s, along with the chromosomes themselves, are segregated (through not positioned) (*Jun and Wright, 2010*). However, we expected that looping of *ori*s at the quarter positions would accelerate this separation. Indeed, we found this to be the case. Looping had a positive effect on *ori* segregation (*Figure 5d*): the greater the looping, the faster *ori* were segregated. Furthermore, the *ori* are not just segregated but are positioned by the spatially varying looping probability to opposite quarter positions along the long axis of the cell in the same way as for the single chromosome case (*Figure 5e*). Similarly, we found that the *ori*s experience an effective restoring force around their respective quarter positions (*Figure 5f*, *Figure 5—figure supplement 1*). The strength of this attraction (the slope of curve) increased with the frequency of looping. Note that unlike our stochastic simulations, we do not need to add repulsion between duplicated *ori*s – entropic repulsion is a natural consequence of the polymer dynamics. Overall, these results indicate that a spatially-varying probability for *ori* to forms loops with other DNA (nominally due to the localised action of MukBEF) leads, in the manner of a DNA relay, to directed movement of *ori*s to the locations where the probability is greatest, i.e. to the locations of MukBEF foci and, furthermore, that this can accelerate *ori* segregation.

## Discussion

In this work, we have presented a quantitative explanation for positioning of the chromosomal origin of replication in *E. coli*. By analysing the positioning and dynamics of *ori* and MukBEF foci in wild-type cells (*Figure 1*), we first showed that *ori* are attracted towards MukBEF foci, as has been previously suggested (*Badrinarayanan et al., 2012a*; *Nicolas et al., 2014*). We have recently argued that the positioning of MukBEF foci can be explained by a stochastic Turing and flux-balance mechanism (*Murray and Sourjik, 2017*). Here, we incorporated *ori* and its interaction with MukBEF into this model and showed how self-organised MukBEF can position origins to their observed mid-cell and quarter-cell positions.

To formulate the model, we needed to specify a particular ansatz for the nature of MukBEF-*ori* relationship. Motivated by SMC in other bacteria (*Gruber, 2018*) and our previous computational results (*Murray and Sourjik, 2017*), we assumed that MukBEF is preferentially loaded onto the DNA at sites within the *ori*. We found that the resultant feedback from *ori* to MukBEF led to robust *ori* partitioning. Preferential loading stabilises the desirable quarter-positioned configuration, preventing stochastic switching to the undesirable configuration having both *ori*s associated to the same MukBEF focus (*Figure 6a*). We describe in more detail how this occurs in *Figure 6—figure supplement 1*. In essence, preferential loading leads to a non-trivial mutual attraction between MukBEF and *ori* that results in robust association, positioning and partitioning of *ori*s as an emergent property. We determined the *ori* drift and diffusion rates by fitting to the experimental *ori* velocity profiles (*Figure 2f*). This also lead to excellent agreement with other experimental measurements that were not used in the fitting, namely the distributions of *ori* positions (*Figure 2e*) and the MukBEF-*ori* separation distance (*Figure 3e*, *Figure 2—figure supplement 4b*), thereby providing further quantitative support for the model. Additionally, we found evidence of the mutual attraction between MukBEF foci and *ori*. MukBEF foci were found to be attracted to *ori* to a similar degree as their

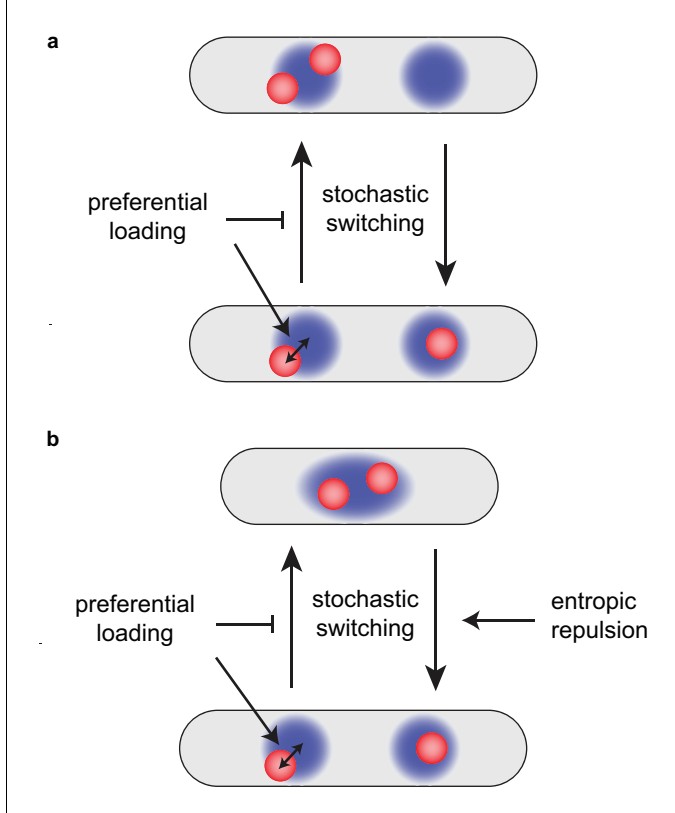

**Figure 6.** Preferential loading and entropic repulsion together lead to the observed *ori* dynamics. (**a**) Schematic illustrating the effect of preferential loading of MukBEF at *ori* in the simulations of non-growing cells. In the presence of preferential loading *ori* (red) and MukBEF foci (blue) are strongly associated with both each other and the quarter positions. This acts to stabilise the desirable correctly partitioned configuration (bottom) over the un-partitioned one (top). In the absence of preferential loading, both configurations have equal stability (are equally likely). See also *Figure 6—figure supplement 1*. (**b**) Short-range entropic repulsion promotes the timely separation of newly duplicated *ori*. Their separation promotes splitting of MukBEF foci. MukBEF-*ori* then move together to opposite quarter positions with preferential loading promoting their association and the stability of the quarter-positioned configuration.

DOI: https://doi.org/10.7554/eLife.46564.020

The following figure supplement is available for figure 6:

**Figure supplement 1.** Schematic of how preferential loading leads to partitioning.
DOI: https://doi.org/10.7554/eLife.46564.021

attraction to mid-cell (*Figure 1e*), consistent with what we observed in our simulations (*Figure 2—figure supplement 5*).

The model could also reproduce the correct *ori* dynamics of a growing cell. This required adding entropic effects (*Jun and Wright, 2010*) to the model. Preferential loading and short-range entropic repulsion are individually not sufficient for both timely and accurate segregation and positioning. However, combined, they give very good agreement with the observed dynamics (*Figure 4*). The repulsion between newly duplicated *ori*s is needed to push the system out of the undesirable configuration immediately after *ori* duplication but is not required for the existence, stability or high colocalisation of the desirable quarter-positioned state (as was seen in simulations of long cells without repulsion) (*Figure 6b*). These properties are the result of preferential loading and the mutual interaction between *ori*s and self-organising MukBEF foci.

Supported by our experimental results, the model assumes that *ori* moves up the MukBEF gradient. What is the physics underlying this biased movement? Since MukBEF can bridge distant regions of the chromosome, it is conceivable that the MukBEF-*ori* relationship, however it is mediated, leads to a higher probability for MukBEF to form bridges between *ori* and other regions of DNA than for other genetic loci. Using polymer simulations, we showed that, combined with the elastic fluctuations of DNA, this can result in directed movement of *ori* up the self-organised MukBEF gradient (*Figure 5*), similar to the DNA relay model (*Lim et al., 2014*; *Surovtsev et al., 2016*) proposed for ParABS-based positioning. However, the situation here is different in that the protein gradient is not generated entirely by the *ori* itself (partition complex in the case of ParABS). In this sense, it is similar to the proposed bulk segregation of chromosomes by membrane-based protein gradients (*Di Ventura et al., 2013*). The proposed mechanism leads to directed movement of *ori*s to the positions of greatest looping (bridging) probability - the middle or quarter-cell positions according to the distribution of MukBEF, as well as accelerated entropic segregation of duplicated *ori*.

It is interesting to compare our results to a previous study of how macro-domain formation and positioning affect chromosome organisation (*Junier et al., 2014*). It was found that a macrodomain formed by spatially *independent* condensation of the *ori* region led to it being pushed to the poles of the cell. The authors therefore needed to additionally impose the mid-cell localisation of the *ori* macrodomain. In our case, the mid-cell location is marked by MukBEF and the increased looping that it induces suffices to keep the *ori* region at that location. Since MukBEF foci are self-positioned (as explained by our stochastic model), no external determinants of location are imposed.

## Predictions

Our model assumes that MukBEF is loaded onto the chromosome at positions within the *ori* region. However, there are other plausible hypotheses for the MukBEF-*ori* relationship. In general, we expect that any relationship that induces a mutual attraction between MukBEF foci and *ori* would result in similar dynamics. Indeed, initial simulations have indicated one possibility is that the *ori* region acts as a 'stop' site for translocating MukBEF complexes. Therefore, the fundamental prediction of our model is not necessarily that MukBEF is loaded at sites within the *ori* region, as for SMC, but rather that whatever the nature of the MukBEF-*ori* relationship, it is such that it leads to an effective mutual attraction between *ori* and MukBEF foci.

In any case, the specificity of *ori* must be specified, directly or indirectly, by some sequence (or sequences) within the *ori* region. We know that the actual site of replication initiation, *oriC*, is not responsible because moving it to another location on the chromosome does not affect *ori* positioning (*Wang et al., 2011*). If this unknown 'centromeric' sequence were inserted into a plasmid lacking a partitioning system, then we would predict that the resulting plasmids would colocalise with MukBEF foci just like the *ori*, and thereby be maintained in the absence of their own partitioning system. However, the challenge lies in identifying the centromeric site as it may be some distance (tens of kb) from *oriC* as is the case for the *parS* sites in *B. subtilis*.

The focus of this work was on *ori* positioning in slow-growing cells. However, an important question and one about which has received comparably little attention, is how *ori* are positioned during faster growth in which cells have multiple replication forks. Youngren et al. have examined the positioning of several genetic loci for the case of four replication forks, i.e. two to four *ori* (*Youngren et al., 2014*). They found that cells are born with two quarter positioned *ori*, that after replication, move to the quarter positions of each cell half. While polymeric effects and bulk chromosome segregation likely play an important role in this behaviour, we nonetheless wondered whether

our simplified model could recapitulate these results. Taking into account the higher copy number of MukBEF in faster growing cells (*Li et al., 2014*), we found very consistent *ori* dynamics (*Figure 7*, *Figure 7—figure supplement 1*). At birth, the two *ori* are quarter positioned, while after replication they migrate to the quarter positions of each cell half. Similar to the slow growth case, we observed that *ori* and MukBEF remain in tight association and MukBEF splitting is coincident with *ori* separation, so that there are approximately as many MukBEF foci as *ori*. This behaviour requires the aforementioned higher copy number. Without it, the number of MukBEF is not substantially different from the slow-growing case and hence correct positioning was not observed. These predictions could be tested in the future by examining MukBEF in these cells as well as the effect of modulating MukBEF expression on the number of foci and on *ori* positioning.

## Outlook

Overall the agreement with the experimental data is very promising given the simplistic nature of the model and that we did not perform a systemic fitting of the parameters to the experimental data (we fit only the *ori*-related parameters – see methods). Nevertheless the depth of the comparison is beyond what has been achieved previously for origin positioning in other bacterial systems. Hence, we suggest that this approach warrants further consideration and that protein self-organisation may have an unappreciated role in chromosome organisation.

More generally, the idea of dynamically controlling the positioning and splitting of a Turing pattern is interesting from a mathematical point of view and may be applicable to other unrelated systems. Indeed, a major aspect of the 'robustness problem' of Turing patterning is the sensitivity of splitting to model parameters, domain size and stochastic effects (*Maini et al., 2012*). The non-trivial coupling to *ori* in our model, as well as the self-positioning nature of the pattern (*Murray and Sourjik, 2017*), goes some way towards mitigating this sensitivity.

As noted earlier, the cubic reaction present in our model leads to a compartment-size effect in dimensions higher than one. One way to overcome this limitation would be to use particle-based

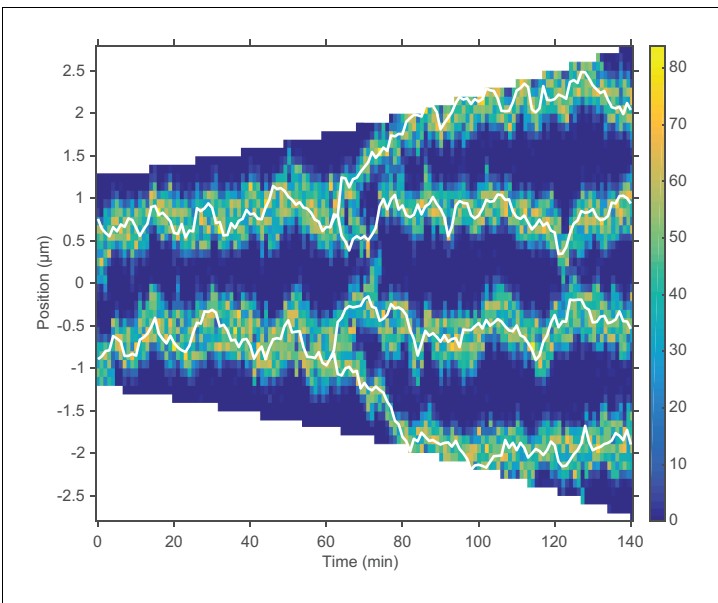

**Figure 7.** The model qualitatively reproduces the observed *ori* dynamics during multi-fork replication. An example kymograph showing multi-fork replication. See also *Figure 7—figure supplement 1*. Results qualitatively agree with *Youngren et al. (2014)*. All parameters are as in *Figure 4*, except for the total number of MukBEF molecules, which was increased from 520 nM to 1000 nM, broadly in line with previous measurements (*Li et al., 2014*).
DOI: https://doi.org/10.7554/eLife.46564.022

The following figure supplement is available for figure 7:

**Figure supplement 1.** Histograms of *ori* positions during multi-fork replication.
DOI: https://doi.org/10.7554/eLife.46564.023

simulation methods, which have recently been extended to higher order reactions (*Flegg, 2016*). However, this would likely involve a substantial increase in computation time, which would make it more challenging to perform multiple runs and parameter sweeps like we have done here. It may rather be possible to reformulate the model in terms of only bimolecular reactions. This would not only be more chemically realistic only but would also allow for higher-dimension lattice based simulations.

For future work, combining particle and polymer simulations, at least to whatever extent is feasible, may provide a deeper understanding of the system. In particular, MukBEF has a major role in chromosome organisation and facilitates long-range DNA interactions (*Lioy et al., 2018*). Like other SMC complexes, MukBEF may act by extruding loops of DNA (*Lioy et al., 2018*), and/or be involved in stabilising them (*Kumar et al., 2017*). Furthermore, MatP, which binds to *matS* sites in the replication terminus region, interacts with MukBEF, displacing it from that region of the chromosome (*Nolivos et al., 2016*) and thereby restricting long-range DNA interactions between the terminus region and other regions of the chromosome (*Lioy et al., 2018*). Both effects may help position this region at mid-cell, while simultaneously encouraging the co-localisation of *ori* with MukBEF (*Nolivos et al., 2016*). Thus, the role of MatP may need to be incorporated into future models. Lastly, with sufficient computing power, a fitting over more model parameters should be possible and may allow the quantitative evaluation of different hypotheses for the MukBEF-*ori* interaction. Overall, we envisage that the study of protein self-organisation in the context of chromosome dynamics has a promising future.

## Materials and methods

### Review of the model

We briefly summarise the underlying model for MukBEF self-organisation (*Murray and Sourjik, 2017*). The general model scheme consists of three 'species' (*Figure 2a*). Species *u* and *v* exist on the surface (the nucleoid), while species *w* exists in the bulk (the cytosol). Species *v* diffuses slower than *u*. In the context of MukBEF, *v* is the basic functional complex, the ATP-bound dimer of dimer, in a state in which it has entrapped multiple strands of DNA and is therefore relatively immobile. Species *u* represents the dimer of dimers in state in which it is non-specifically associated to DNA but without being immobilised e.g. it has not entrapped any DNA strands. These two species interconvert with the latter becoming the former at a basal rate α and cooperatively at a rate β, while the former becomes the latter at a rate γ. The cytosolic state *w* is the ATP-unbound dimer state which converts to and from the DNA associated states with linear rates ε and δ respectively. The differential equations describing the model are given in *Figure 2—figure supplement 2a*. Parameter values are specified below.

This system exhibits Turing pattern formation i.e. the differing diffusion rates and the reactions of the system are such that a diffusion-driven instability can occur that leads to the spontaneous formation of spatially varying concentration profiles. The two states *u* and *v* generate the Turing pattern while the cytosolic state *w* is responsible for positioning the pattern. The latter is required to be well-mixed and it is so because it interconverts with the Turing species *u* and *v* on a sufficiently slow timescale. Note that more generally this state is not required to be cytosolic, only well-mixed. The kymographs and distributions shown in this work are of *v* (which we simply refer to as MukBEF). See *Murray and Sourjik (2017)* for further details and a detailed justification of the model.

### Stochastic simulations

Stochastic simulations were performed in C++. We used the Gillespie method (also called the Stochastic Simulation Algorithm) (*Gillespie, 2007*; *Gillespie et al., 2013*) to obtain exact realisations of the Reaction Diffusion Master Equation (RDME) as described previously (*Murray and Sourjik, 2017*) but with some changes for efficiency and the addition of simulated *ori*. We replaced the binary tree search of the enhanced direct method (*Gibson and Bruck, 2000*) with a 2D search as proposed by Mauch and Stalzer (*Mauch and Stalzer, 2011*) and switched from 32-bit uniform random numbers (using the Ziggurat method) to 64-bit numbers (using std::mt19937_64) to ensure enough significant digits to accurately sample reactions occurring with very low relative rates (namely, *ori* diffusion). As before, the spatial domain (the long axis of the cell) is divided into discrete compartments, each

having a width of $h = 0.1$ m and between which the species can diffuse (*Figure 2c*). The cytosolic state is well-mixed and is therefore treated implicitly. The system state was read out every 60s for consistency with the experimental procedure. For simulations with growth, the simulation was paused after every time-duration that corresponded to growth by one compartment. An additional (empty) compartment was then inserted at a random position and the volume and total number of molecules (via the cytosolic fraction) were increased, maintaining the same overall concentration.

The simulations were extended from those of the previous work by the addition of *ori*. We treated the *ori* as an additional diffusing species (with only one to four copies as appropriate) and implemented its biased diffusion up the MukBEF gradient, its duplication and its repulsion from other *ori* as follows.

## Diffusion of *ori* subject to the MukBEF gradient

We assume that MukBEF is linearly related to the potential surface experienced by *ori*. This is the simplest choice and is supported by the agreement of the resultant linear velocity profile with the experimental one. Furthermore, since any symmetric potential is approximately quadratic around its minimum (up to third order due to symmetry), we would in any case expect a linear velocity profile and since we have the best statistics near the minimum (the MukBEF peak), we would likely not be able distinguish other relationships between the MukBEF concentration and the potential it generates.

Given a linear relationship, the drift *ori* experience is based on the derivative of the local MukBEF concentration. We use jump rates (the rate at which *ori* jump between neighbouring compartments, illustrated in *Figure 2c*) derived by Wang, Peskin and Elston (*Wang et al., 2003*). The forward and backward jump rates from compartment $i$ to $i + 1$ and $i − 1$ respectively are

$$F_{i,i+1} = \frac{D_{ori}}{h^2} \frac{\alpha_{i,i+1}}{1 - e^{-\alpha_{i,i+1}}}$$

$$B_{i,i-1} = \frac{D_{ori}}{h^2} \frac{\alpha_{i,i-1}}{e^{\alpha_{i,i-1}} - 1}$$

where the dimensionless quantity $\alpha_{i,j} = \frac{\mu(v_j - v_i)}{h}$ is, up to a factor, the difference in the MukBEF concentration between the compartments ($v_i$ is the number of molecules of slowly diffusing MukBEF species in compartment $i$), $D_{ori}$ is the diffusion constant and $\mu$ is the drift parameter determining the strength of the attraction up the MukBEF gradient. We use the difference in the slowly-diffusing species only as MukBEF clusters have been shown in vivo to consist only of this state (*Badrinarayanan et al., 2012b*). This form for the jump rates respects detailed balance since the exchange between two neighbouring compartments balances i.e. $F_{i,i+1} = B_{i+1,i}$. The derivation of these rates relies on the assumptions that, within individual compartments, the probability density for *ori* is at steady state and that the MukBEF gradient is approximately linear. Both of these requirements can be satisfied for sufficiently small compartment widths. However, it is not feasible to decrease the compartment width much below 0.1 μm due to the increased computationally cost. Yet, the often sharp MukBEF profile (at a fixed moment in time) suggested that shorter compartment widths might be required. We therefore introduced sub-compartments within every compartment but only for *ori* positions. This approach has previously been applied to stochastic Turing patterns (*Cao and Erban, 2014*) but here we apply it to a 'non-Turing' species (*ori*). Each compartment was divided into an odd number of sub-compartments and the MukBEF concentration was linearly interpolated across sub-compartments. The jump rates between sub-compartments were then defined as above. Performing simulations for different numbers of sub-compartments, we found that the apparent diffusion constant and drift rate (see below and *Figure 2*) stabilised with greater than approximately 5 sub-compartments. The apparent diffusion was approximately 40% higher without sub-compartments. Since higher numbers of sub-compartments carried very little computational cost, we chose an arbitrary but relatively high value of 21 sub-compartments for the simulations presented in this work in order to be confident that there are no sub-compartment-size dependent effects.

## Duplication of *ori*

The timing of *ori* replication was chosen randomly in each simulation by picking a duplication length from a normal distribution with mean 3 m and coefficient of variation 0.16 (based on the distribution of *ori*-foci splitting length of the data in *Kuwada et al., 2013*). As we use a fixed growth range (2.5-5 μm) and doubling time (120 min), we truncate the distribution to this range. This duplication length is then converted to a duplication time via the exponential relationship between cell length and time. The simulation is paused when it reaches this time, the *ori* is duplicated (remaining within the same sub-compartment) and the simulation continued. Note that we do not mimic cohesion of newly replicated stands so that what we refer to in the simulation as *ori* duplication actually more closely corresponds to initial *ori* separation in vivo, which occurs 10-15 minutes after replication initiation.

## *ori* repulsion

As discussed in the text, newly duplicated *ori* are likely to experience, for entropic reasons, a repulsive force between them (*Jun and Mulder, 2006*). Numerical studies have shown that the corresponding potential has the qualitative form of a Gaussian in the centre-of-mass separation (*Bohn and Heermann, 2011*; *Bohn and Heermann, 2010a*). We assume that we are in the overdamped regime such that the separation velocity $v_s$ due to this force is proportional to the force. We therefore have the form $v_s = k\, d\, e^{-\frac{1}{2}\left(\frac{d}{\sigma}\right)^2}$, where d is the separation between *ori*, σ is the range and k is the strength.

## Parameters

All parameters of the core MukBEF model are given in *Table 1* and are as previously described and justified (*Murray and Sourjik, 2017*), except for the total species concentration C, which is increased by 30% to 520 nM but which is still within the experimentally justified range (*Badrinarayanan et al., 2012b*; *Li et al., 2014*). This was done for compatibility of the MukBEF splitting time with the lower range of cell lengths used in this work (2.5 μm – 5 μm), which were chosen to more closely match the range of the experimental data in *Kuwada et al. (2013)*. The cell volume (V = $1.25 \times 10^{-15}$ L at birth (2.5 μm)) was taken to scale linearly with length and is required to convert the parameter β to the appropriate dimensions for use in the stochastic simulations. With the above total concentration and cell volume, there are 391 simulated molecules at birth. The remaining (*ori*-related) parameters were chosen by comparison with experimental data as described below and are given in *Table 2*, *Table 3*, *Table 4* and *Table 5*.

## Initialisation of simulations

Initial concentrations were set to the integer homogenous configuration closest to the deterministic homogeneous state. Unless stated otherwise, single *ori* were initially placed at mid-cell, while in simulations starting with two *ori,* they were placed at the quarter positions. Simulations were first run for 30 min to equilibrate and then read out every 1 min (chosen to match the experimental data).

**Table 1.** Common Parameters.

| Parameter | Value |
| --- | --- |
| α | 0.5 s$^{-1}$ |
| β | $1.5 \times 10^{-4}$ nM$^{-2}$ s$^{-1}$ |
| γ | 3.6 s$^{-1}$ |
| δ | log(2)/50 s$^{-1}$ |
| ε | 3δ |
| $D_u$ | 0.3 μm$^2$ s$^{-1}$ |
| $D_v$ | 0.012 μm$^2$ s$^{-1}$ |
| V (volume at length 2.5 μm) | $1.25 \times 10^{-15}$ L |
| C | 520 nM |

DOI: https://doi.org/10.7554/eLife.46564.024

**Table 2.** Additional parameters used in *Figure 2d*, *Figure 2—figure supplements 1–4*, *Figure 3*, *Figure 3—figure supplement 1*.
Obtained by fitting the model without preferential loading to the data of Kuwada et al. Some simulations use preferential loading as stated in the legend or on the plot axis.

| Parameter | Value |
| --- | --- |
| $D_{ori}$ | $5.4 \times 10^{-5}$ µm$^2$ s$^{-1}$ |
| $\mu$ | 0.026 µm |

DOI: https://doi.org/10.7554/eLife.46564.025

## Apparent *ori* diffusion constant and drift rate

To be able to quantitatively compare the experimental and simulated data, we needed quantitative descriptors of the *ori* dynamics. We compared both data sets to a theoretical model of particle diffusion in a harmonic potential $U = \frac{1}{2}fx^2$ over an infinite 1D domain. Given a particle at position $x_0$, the probability density that it is at position $x$ at time $\delta t$ later is (*Doi and Edwards, 1988*)

$$p(x, \delta t | x_0) = \sqrt{\frac{f/kT}{2\pi S}} exp\left[ -\frac{f/kT}{2S}\left( x - x_0 e^{-\delta t/\tau} \right)^2 \right],$$

where $S = 1 - e^{-2\delta t/\tau}$ and $\tau = \frac{kT}{fD}$. From this, it is straightforward to calculate the expected value and variance of the step-wise velocity $v := \frac{x-x_0}{\delta t}$:

$$E[v] = \frac{e^{\frac{-\delta t}{\tau}} - 1}{\delta t} x_0 \approx -\frac{x_0}{\tau}$$

$$Var(v) = \frac{D\tau}{\delta t^2}\left( 1 - e^{-2\delta t/\tau} \right) \approx \frac{2D}{\delta t},$$

where the second equality holds for $2\delta t/\tau \ll 1$ (the full expression is used when fitting). Note the expected value of the step-wise velocity depends linearly on position, while the variance is independent of position. This is observed in both experiments and simulations close within the neighbourhood of the *ori* 'home' position. We therefore use the measured slope of the velocity relationship and its variance to determine an apparent diffusion constant $D_a$ and drift rate $d_a = \frac{1}{\tau} = \frac{fD_a}{kT}$. Linear fitting (*Figure 2*, *Figure 2—figure supplement 3*) was performed using the *fit* function in Matlab with the inverse square of the standard errors as weights.

## *ori* drift and diffusion parameters

To search for parameter values for the *ori* diffusion constant ($D_{ori}$) and the strength of attraction towards MukBEF () that gave agreement between the measured apparent diffusion constants ($D_a$) and drift rates ($d_a$), we performed simple parameters sweeps. For the initial fitting (*Figure 2—figure supplement 3*), we chose $D_{ori}$ to be a percentage of the experimentally measured diffusion constant $D_a$, ranging from 70% to 110% in 5% intervals, while the drift parameter was ranged from 0.5 to 2.5 times a nominal value of 0.026 µm (in steps of 0.5). The combination giving the best agreement and shown in *Figure 2—figure supplement 3* was $D_{ori}$ = 0.9 $D_{ori}$ = 5.4 × 10$^{-5}$ µm$^2$s$^{-1}$ and $\mu$ = 0.026 µm. These values were also used for the simulations shown in *Figure 2d*, *Figure 2—figure supplements 1–5*, *Figure 3* and *Figure 3—figure supplement 1*.

**Table 3.** Additional parameters used in *Figure 2e,f*, *Figure 4*, *Figure 4—figure supplement 1–3*.
Obtained by fitting the model with 6X preferential loading to the data of Kuwada et al. (*Figure 2e,f*).

| Parameter | Value |
| --- | --- |
| $D_{ori}$ | $5.1 \times 10^{-5}$ µm$^2$ s$^{-1}$ |
| $\mu$ | 0.052 µm |

DOI: https://doi.org/10.7554/eLife.46564.026

**Table 4.** Additional parameters (together with those in **Table 3**) used in **Figure 4—figure supplement 3**.

*ori* repulsion without preferential loading.

| Parameter | Value |
| --- | --- |
| $k$ | 5 s$^{-1}$ |
| $\sigma$ | 200 nm |

DOI: https://doi.org/10.7554/eLife.46564.027

To produce **Figure 2e, f**, we performed the same parameter sweep in the presence of 6x preferential loading and found that the best agreement was obtained with $D_{ori} = 0.85 \, D_a = 5.1 \times 10^{-5} \, \mu m^2 s^{-1}$ and $\mu = 2 \times 0.026 \, \mu m$. It should be noted that given the stochastic nature of the simulations, even with 100 runs of 600 min each, there was quite some variability in the data. These parameters were used for subsequent simulations with growth.

## Entropic repulsion of *ori*

We performed a parameter sweep of the strength and range of the *ori* repulsion and the preferential loading ratio (**Figure 4—figure supplement 2**). The range σ was varied over the values 50, 100, 200, 300, 400 nm, while the strength $k$ was varied over 0.2, 1, 5, 25, 125 s−1. We performed 450 independent simulations of a growing cell as described and measured the partitioning accuracy and relative separation of *ori* 20 min after *ori* duplication. We repeated this for different preferential loading ratios. The result was **Figure 4—figure supplement 2**.

## Polymer simulations

In order to investigate the interplay between MukBEF and *ori* positioning within the nucleoid, we used a coarse-grained lattice polymer (**Bohn and Heermann, 2010b**). Within this framework, the DNA is described as a self-avoiding ring polymer that is confined in an elongated cuboid with an aspect ratio of 4:1 comparable to that of the *E. coli* nucleoid. Using a ring polymer composed of 464 monomers, we chose a lattice of size 22 × 22 × 88 that leads to a system density (monomer-to-volume ratio) of around 10%. Dynamic looping interactions were enabled between one specific monomer (*ori*) and distant monomers. For simplicity, we did not include loop formation between any two arbitrary sites but we do not expect this to change the nature of our results other than giving a homogeneous background of looping events. The looping probability is set to be dependent on the spatial position of *ori* along the long axis of the nucleoid and is drawn from a Gaussian distribution centred around mid-cell with a standard deviation of 8.8 lattice units. A lifetime of 10000 Monte-Carlo steps (MCS) was assigned to each loop. 50 independent Monte-Carlo trajectories were used to sample the dynamics of the system. In each simulation, 10000 polymer conformations were recorded, one every 50000 MCS. The initial position of *ori* was varied in each simulation in order to uniformly cover the long axis of the cuboid.

In the simulations of the two chromosomes, each chromosome was modelled as a ring polymer composed of 232 monomers in a cuboid of the same size. Hence, the system density of 10% stayed the same compared to the single-chromosome simulations. The spatially-varying probability for looping between both either of the two *ori*s and a distant monomers of any of the two polymers were drawn from the superposition of two Gaussian distributions centred around the two quarter positions of the cuboid with a standard deviation of 8.8 lattice units. 500 independent Monte-Carlo

**Table 5.** Additional parameters (together with those in **Table 3**) used in **Figure 4**.

*ori* repulsion with 10x preferential loading.

| Parameter | Value |
| --- | --- |
| $k$ | 1 s$^{-1}$ |
| $\sigma$ | 200 nm |

DOI: https://doi.org/10.7554/eLife.46564.028

trajectories were used to sample the dynamics of the two polymer system. The simulations were initialized with two overlapping polymers with the two *ori*s at the centre of the cuboid.

## Experiments

Strain SN192 (AB1157 *lacO240-hyg* at *ori1*, *tetO240-gen* at *ter3, Plac-lacI-mCherry-frt* at *leuB, Plac-tetR-mCerulean-frt* at *galK, mukB-mYPet-frt*) (*Nolivos et al., 2016*) was grown in M9 minimal medium supplemented with 0.2% glycerol and required amino acids (threonine, leucine, proline, histidine and arginine—0.1 mg ml$^{-1}$) at 30°C. Cells were grown O/N, diluted 1000-fold and grown to an $A_{600}$ of 0.05–0.2. Unlike longer cells with two (quarter positioned) *ori* foci, cells with a single (mid-cell localised) *ori* focus, can be analysed together in absolute, rather than scaled, coordinates by simply measuring foci positions relative to mid-cell (as we did in *Figure 2* for the dataset of *Kuwada et al., 2013*). We therefore, unless otherwise indicated, treated cells with DL serine hydroxamate (SHX) (Sigma-Aldrich, S4503) to a final concentration of 1 mg/ml. During the treatment, cells do not initiate a new round of replication, but complete any ongoing rounds (*Ferullo et al., 2009*). To allow sufficient time for replication to complete to termination, cultures were grown for 3 hr in the presence of SHX (generation time ~170 min). Finally, cells were spotted onto an M9-glycerol 1% agarose pad with the growth medium on a slide for imaging.

Time-lapse movies were acquired on a Nikon Ti-E inverted fluorescence microscope equipped with a perfect focus system, a 100 × NA 1.4 oil immersion objective, an sCMOS camera (Hamamatsu Flash 4), a motorised stage, and a 30°C temperature chamber (Okolabs). Fluorescence images were automatically collected at 1 min intervals for 56 min using NIS-Elements software (Nikon) and an LED excitation source (Lumencor SpectraX). Exposure times were 150 ms for mCherry, and 100 ms for mYPet using 50% LED intensity. Phase contrast images were also collected at 1 min intervals for cell segmentation.

Alignment of frames, cell segmentation and linking of cells in consecutive frames were performed using SuperSegger (*Stylianidou et al., 2016*). To ensure that we considered only cells with a single *ori*, we subsequently filtered the dataset as follows. For any cells that had two *ori* foci on any frame, we kept only the frames before two foci were first detected. This reduced the dataset of SHX treated from 1431 to 952 cells. We then used only frames with exactly one *ori* focus and one MukB focus. This resulted in 31820 data points (cell-frame combinations). Similar results were obtained if we further restricted the data set to cells of similar length. The same data was used to generate the step-wise velocity profiles (*Figure 1d,e*) but as two consecutive frames are required this reduced the data set to 26226 data points.

Analysis, fitting and plotting were performed in MATLAB (Mathworks Inc).

## Acknowledgements

We thank Remy Colin for discussions and Victor Sourjik for discussion, support and comments on the manuscript. We also thank Nathan Kuwada and Paul Wiggins for providing the raw data of their previous work. SMM was supported by the German Federal Ministry of Education and Research and the Max Planck Society in the framework of the MaxSynBio research network. AH was supported by a grant from the International Human Frontier Science Program Organization (RGP0014/2014) and the Heidelberg Graduate School of Mathematical and Computational Methods for the Sciences (HGS MathComp), funded by DFG grant GSC 220 in the German Universities Excellence Initiative. The polymer simulations were performed on the bwForCluster supported by the state of Baden-Württemberg through bwHPC and the German Research Foundation (DFG) through grant INST 35/1134–1 FUGG. Work in the Sherratt laboratory was supported by a Wellcome Investigator Award (DJS: 200782/Z/16/Z).

## Additional information

### Funding

| Funder | Grant reference number | Author |
| --- | --- | --- |
| Wellcome | DSJ: 200782/Z/16/Z | Jarno Mäkelä<br>David J Sherratt |

| Deutsche Forschungsge-meinschaft | GSC 220 | Andreas Hofmann |
|---|---|---|
| Max-Planck-Gesellschaft | Open-access funding | Sean M Murray |
| Human Frontier Science Program | RGP0014/2014 | Andreas Hofmann |
| Deutsche Forschungsge-meinschaft | INST 35/1134-1 FUGG | Andreas Hofmann Dieter Heermann |

The funders had no role in study design, data collection and interpretation, or the decision to submit the work for publication.

### Author contributions
Andreas Hofmann, Software, Investigation, Methodology, Writing—review and editing; Jarno Mäkelä, Investigation, Methodology, Writing—review and editing; David J Sherratt, Dieter Heermann, Data curation, Supervision, Funding acquisition, Writing—review and editing; Seán M Murray, Conceptualization, Data curation, Software, Formal analysis, Investigation, Visualization, Methodology, Writing—original draft, Project administration, Writing—review and editing

### Author ORCIDs
Andreas Hofmann (iD) https://orcid.org/0000-0002-4800-8429
Jarno Mäkelä (iD) https://orcid.org/0000-0003-1844-2619
David J Sherratt (iD) http://orcid.org/0000-0002-2104-5430
Seán M Murray (iD) https://orcid.org/0000-0002-2260-0774

### Decision letter and Author response
Decision letter https://doi.org/10.7554/eLife.46564.031
Author response https://doi.org/10.7554/eLife.46564.032

## Additional files

### Supplementary files
• Transparent reporting form
DOI: https://doi.org/10.7554/eLife.46564.029

### Data availability
Experimental source data files have been provided for Figure 1. We also used the ori localisation tracks provided as supplementary data to Kuwada et al. (2014) and the co-localisation curves from Figure 1c of Nolivos et al. (2016).

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
