## [Decision Letter]

Thank you for submitting your article "Self-organised segregation of bacterial chromosomal origins" for consideration by *eLife*. Your article has been reviewed by three peer reviewers, and the evaluation has been overseen by a Reviewing Editor and Naama Barkai as the Senior Editor. The reviewers have opted to remain anonymous.

The reviewers have discussed the reviews with one another and the Reviewing Editor has drafted this decision to help you prepare a revised submission.

Summary:

In the manuscript "Self-organised segregation of bacterial chromosomal origins", the authors investigate the segregation of the chromosomal origins in *Escherichia coli* bacteria. They build on previously published work (Murray and Sourjik, 2017) about the dynamic and self-organized clustering of MukBEF proteins on the chromosome. To study the dynamics of the chromosomal origins they incorporate interactions between MukBEF and the origins into the model, as suggested by experimental findings. Their main result is that preferential loading of MukBEF at the origins and entropic repulsion of the origins can explain the experimentally observed ori dynamics. The presented work is an important contribution to quantitative cell biology because it gives new mechanistic insights into origin positioning in *E. coli* by integration of computational methods with experimental data.

The reviewers were supportive of the approach and the results, but have raised a number of issues that need to be addressed in a revised manuscript in order for the paper to be acceptable for publication in *eLife*. In particular, additional modeling work is requested, along with detailed, falsifiable predictions, and a discussion of the limitations of the current model.

Essential revisions:

1) Limitations of 1D modeling.

The authors' modeling is limited to 1D. While I appreciate the technical challenges, 3D modeling is a conceptual necessity. The population average aspect ratio of *E. coli* is typically length/width = 4, and treating an *E. coli* cell as a 1D space is really misleading for chromosome organization and segregation. Such an 1D approach was acceptable for Min oscillations some 20 years ago when physicists modeled them as a 1D reaction-diffusion problem; not only such approaches were new then, but also the nature of dynamics of Min allowed dimensional reduction as the Min proteins bind and unbind to the inner cell-wall membrane, which was key to Min oscillations. However, that is not the case for MukBEF. See also below the points on the polymeric nature of the chromosome.

2) 3D effects and robustness.

*E. coli* cells with mild inhibition of mreB expression become almost completely round, but they still grow at the same growth rate as wildtype showing multifork replication (e.g., PMID: 28416114). We are unsure how ori positioning (if at all) by MukBEF pattern formation is important for chromosome segregation in round cells. Please explain and make predictions.

3) Polymeric nature of the chromosome.

Another aspect of the chromosome the authors almost completely neglect is the polymeric nature of the problem. Ori, after all, is an extremely small fraction of the chromosome. How does the ori positioning explain the principal organization of the rest of the chromosome? The authors did add limited 3D polymeric simulations, but they are far from being sufficient to justify the main claim of the paper.

4) Ori dynamics during (non-overlapping) replication.

In the last part of the manuscript, the authors do consider the polymeric nature of the chromosome and model the gradient of MukBEF implicitly by dynamic looping model with only one ori. They show that the ori can have a directed motion towards the cell center where the looping probability is more, but it is not clear how the oris reach the quarter positions during replication using the dynamic looping model. How does the dynamic looping model explain the position of MukBEF at quarter positions?

5) Ori dynamics during multifork replication.

We cannot assess the robustness of preferential loading of MukBEF on ori segregation and positioning during multifork replications (when 2 or 3 replication cycles overlap with up to 8 oris). This must be one of the most straightforward predictions the authors should be able to make based on their model, and they should extend their work to multifork replication cases. To be clear, we are not asking for more experiments, just falsifiable predictions using their model for future experimental tests.

6) Predictions.

In our view, modeling would be far more informative when it makes falsifiable predictions, rather than when the ad-hoc assumptions can produce results that look like published data. We ask the authors make two predictions, and include them in their revised manuscript.

Prediction 1: what will happen to low-copy number plasmids without partitioning system, when they instead integrated the *E. coli* chromosome ori? We presume the authors would predict these plasmids would completely localize with MukBEF clusters given that entropic repulsion would be negligible. If validated, this prediction would significantly support both the authors MukBEF model as well as the role of entropic repulsion.

Prediction 2: what would happen to the ectopic ori in the chromosome, depending on its locus position in the chromosome? And what would happen to the ectopic ori during multifork replication? Such strains were constructed previously and tested in limited growth conditions (PMID:21670292), and it would be worthwhile to extend the authors work by making explicit predictions assuming various different locations of the ectopic ori under different growth conditions.

See also other suggested predictions above.

7) The model is based on the known flux-balance mechanism and extended to explain the movement of the chromosomal origins. With each experimental observation (separation of two oris, ori dynamics during growth) they add a new aspect to the model (preferential loading, repulsive interaction of oris), which then leads to good agreement of the simulation results with the data. This raises two important questions: a) Are there new insights the model provides beyond the data? b) Are the model assumptions justified? In particular, is a preferential loading ratio of six, which shows the smallest variance of the peaks, realistic? The authors refer to a measured preferential loading for SMC in *B. subtilis*. What is the preferential loading ratio here? Is it possible to get a measurement for MukBEF in *E. coli*? Furthermore, is the chosen repulsive interaction strength / range realistic? It might be that, although repulsion contributes to the dynamics of the origins in the in vivo system, it is not the main factor because it might not be strong enough. The values for the repulsive strength and range are obtained by fitting (Figure 4——figure supplement 2). Isn't it trivial that with this fitted choice of parameters the experimental observations are reproduced?

8) In the model, diffusion of the origins is biased by a gradient of MukBEF. This is an important assumption of the model, which raises several questions: What is the physics behind this assumption, i.e. where does the net force come from? What determines the strength of the bias? Can the strength be measured? Another assumption of the model is that only the concentration of the slow species (in blue in Figure 2A) is considered for the bias of the origins. Why are not both, the slow and the fast moving, nucleoid-bound species (in green in Figure 2A) considered for the bias? Couldn't the success of the model in explaining the data hinge to a large degree on all these assumptions?

9) The authors justify their assumption of a one-dimensional model with the fact that most of the foci are within 40% of cell width and "There are also technical reason inherent to the method that make nonlinear models in higher dimensions (and the corresponding lower voxel volumes) problematic." However, 40% of cell width is not a very confined region. Furthermore, could the authors state more clearly, which technical reasons they refer to? If it is the lower bound on the mesh size, there are generalized RDMEs that overcome this limit (Hellander and Petzold, 2016).

10) Most of the parameters for the MukBEF dynamics are chosen as previously described (Murray and Sourjik, 2017). This reference should be cited when the parameters are discussed (subsection “Parameters”). The only parameter that is chosen differently is the concentration of MukBEF, which is set to a larger value than the one used before. Why is this choice justified? Couldn't the concentration be measured experimentally? Since the strength of stochastic effects depends on the number of particles, it is crucial to choose realistic concentrations / number of particles as well as realistic rates in the model.

11) The authors did a number of experiments with *E. coli* cells, in which they imaged the ori location(s) and/or Muk-foci. Together with what is already known in the literature, these experiments constitute the empirical basis for the analysis of the authors. However, instead of presenting this empirical basis in a coherent fashion in the beginning of the Results section, the authors chose to present the results piecemeal. In particular, the results that we found most striking, the "restoring velocities" of Figure 5B and D, are shown very late in the paper and presented only as a confirmation of their hypothesis of preferential Muk-loading onto the ori locus. In our view, this data should be the motivation for the hypothesis – prior to seeing this data we kept wondering what the basis for the hypothesis was.

Then, after finally seeing the "restoring velocities" of Figure 5B and D, we thought that the most logical next step would be to repeat this analysis on different variants of the MukBEF-system, in order to test whether the higher restoring velocity towards MukB than to midcell is lost when the system is modified in one or the other way.

Instead, the authors continued by presenting computer simulations of chromosome dynamics within a dynamic loop model. The presentation was so brief and the analysis of the simulations so minimal that the conclusions from this part of the work remained obscure to us. We think these simulations were meant to explore whether the directed motion of ori (within their model) is due to a real mechanical force on ori (and, if so, where this force comes from), or whether it results from a ratchet-like mechanism. However, what is the conclusion on this question and how do the authors arrive at the conclusion? We think this part of the work should either be eliminated or should be elaborated on and presented much more clearly.

12) The logic underlying the simulations of the stochastic model of Figure 2C was much clearer. However, given that this model relies heavily on the prior work of Murray and Sourjik, 2017, it would be good to spell out more clearly what has already been established in that paper and what is added here. Furthermore, the way in which the comparison of these simulations with the experimental data is done is sometimes confusing, in the sense that we did not get a clear sense about "what is put in and what they got out". For instance, it says "… Note this was based on a direct comparison and not a fit to the experimental distribution.", while earlier, it says "… We found that by adjusting the diffusion and drift input parameters, we were able to obtain excellent agreement with the experimental results (Figure 2F)." (the latter indicates fitting)

In conclusion, we recommend that the authors rewrite their manuscript to present the data and the logic in a more coherent and stringent way. For instance, we think it is in the best interest of the authors to clearly point out what results of this paper remain true in case future studies should show that there is no preferential loading of Muk onto ori. Our impression is that the data of Figure 5B and D is an important part of what would survive in that case. It would be helpful, if the authors clearly spell out what can be concluded from this data independent of a specific modeling scenario. It would be even better to also add data that tests the restoring velocities of variants of the Muk system.

---

## [Author Response]

Essential revisions:1) Limitations of 1D modeling.The authors' modeling is limited to 1D. While I appreciate the technical challenges, 3D modeling is a conceptual necessity. The population average aspect ratio of E. coli is typically length/width = 4, and treating an E. coli cell as a 1D space is really misleading for chromosome organization and segregation. Such an 1D approach was acceptable for Min oscillations some 20 years ago when physicists modeled them as a 1D reaction-diffusion problem; not only such approaches were new then, but also the nature of dynamics of Min allowed dimensional reduction as the Min proteins bind and unbind to the inner cell-wall membrane, which was key to Min oscillations. However, that is not the case for MukBEF. See also below the points on the polymeric nature of the chromosome.

Also in response to point #9 below. There are important differences between our study and those of the Min system. The Min system is typically, modelled using a quadratic interaction (bimolecular reaction), whereas our model, as is typical for (static) Turing patterns, has a cubic interaction. As this reviewer has noted there is a fundamental limit on the mesh size (voxel volume) when using non-linear reactions in lattice based Gillespie/Monte Carlo simulations. Several groups have proposed ways of overcoming this problem. However, all such ‘fixes’ are for bi-molecular reactions, including the recent work of Hellander and Petzold. To our knowledge there is currently no way to overcome the problem for tri-molecular reactions. We have now explained this problem in more detail in the text and Materials and methods section and include a quantification of the mesh-size effect (Figure 2—figure supplement 5).

Note that we have previously performed deterministic 3D simulations of the core MukBEF model (Murray and Sourjik, 2017). However, as there are only a few hundred MukB molecules in the cell, and only 1 or 2 ori, the model must be simulated stochastically.

The presence of trimolecular reaction term in models of Turing patterning has always been problematic due to their somewhat unrealistic nature. Attempts have been made to introduce an intermediate species that is in equilibrium with one of the other states and thereby replace one trimolecular reaction with two bimolecular reactions. This can work for some models e.g. the Brusselator (Smith and Dachau 2018, doi.org//10.1098/rsif.2017.0805) but not all depending on the precise form of the reactions (our attempts with the approach have given unsatisfactory results). Furthermore any additional species should be biological justifiable. We therefore think that the solution will come from future experimental results or re-interpretation of existing data, something on which we are actively working.

Another important point to make concerns the comparison to experimental data. To our knowledge, the comparisons of the Min system to experiment have always been qualitative in nature. We on the other hand make quantitative comparisons between simulated and measured ori dynamics. Furthermore, since we only have 1 ori in a short cell, we must necessarily perform many simulations for long times to obtain sufficient statistics (in Figure 2, we perform 100 simulations of 600 min simulation time each). The average distribution of MukBEF we can determine more easily – the ori is the limiting factor statistics-wise. We also perform parameter sweeps, with each parameter set requiring the aforementioned number of simulations. Running in parallel on 20 cores it takes a few days to complete a sweep of only 64 parameter choices. Performing such sweeps for a 3D model could easily multiply this by a factor of several hundred or more (with small mesh-sizes diffusion events become much more frequent and dominate the simulation time) and so even without the issue of mesh-size effects, performing all the analyses and parameter sweeps that we have done in 3D is not a straightforward request.

Furthermore, we would like to point out that many groups have worked on the Min system over many years with many developments on both the experimental and modelling sides. This is our first paper on the ori-MukBEF model and we can therefore hardly be expected to have studied the system to the same degree. We would also like to point out that the Min system is still commonly modelled deterministically (e.g. Halatek and Frey 2018, Nature Physics; Denk et al. 2018, PNAS; Wu et al. 2016, Mol.Sys.Bio.).

Lastly, concerning the reviewer’s comment that MukBEF foci and ori being within the centre 40% (300 nm) of the short axis of the cell does not count as being confined, we now comment on the size of MukBEF (140 nm arc length, 70 nm from the hinge to each head domain). It is also worth noting that we are interested and measure ori and MukBEF foci positioning only along the long axis of the cell and that segregation, by definition, occurs along this axis. Our reaction diffusion simulations are also not single-occupancy: MukBEF molecules and ori can move ‘through’ each other.

2) 3D effects and robustness.E. coli cells with mild inhibition of mreB expression become almost completely round, but they still grow at the same growth rate as wildtype showing multifork replication (e.g., PMID: 28416114). We are unsure how ori positioning (if at all) by MukBEF pattern formation is important for chromosome segregation in round cells. Please explain and make predictions.

The cells in the given reference, while certainly shorter and wider than WT are nonetheless far from round, with an aspect ratio not much lower than WT. Indeed, depletion of MreB or inhibition of MreB structures (using for example the antibiotic A22), results in completely round cells that then lyse (PMID: 15612918). However, the lethality can be suppressed in these cells by overexpressing the cell division genes ftsZAQ. To our knowledge, chromosome organisation in these cells has not been studied in detail. We do not know if these cells grow at the same rate as wildtype nor if ori and MukBEF foci (which presumably still form) are specifically positioned within the cell. There is also evidence that MreB interacts with the topoisomerase TopoIV, thereby linking it to chromosome organisation and segregation (PMID: 19187760). Thus, geometry is likely not the only determining factor for the positioning of ori in these cells. Given that and the fact that we do not explicitly model the chromosome, we would argue that the topic is not in the scope of the paper, which is focused on slowing-growing wild-type cells.

However, ignoring other such effects, we can speculate about what would happen in the context of our model. We have previously showed that geometry sensing by the flux-balance mechanism works in 3 dimensions (within a rectangular cuboid) using the deterministic version of our model (Murray and Sourjik, 2017) and it has recently also been shown to work in 3 dimensions in the context of a stochastic model of PomXYZ positioning in M. xanthus (Biorxiv: https://doi.org/10.1101/496364). We therefore expect that the flux-balance mechanism could find the centre of round cells, and the approximate quarter positions of oblong dividing cells. MukBEF foci would form at these positions and recruit and partition ori to the same locations. This is our speculative prediction. As our stochastic simulations are not reliable in 3D, we cannot verify that this would be the result of our model. Therefore, we have not added this to the revised manuscript.

3) Polymeric nature of the chromosome.Another aspect of the chromosome the authors almost completely neglect is the polymeric nature of the problem.

Our model is a simplified one, out of necessity. It is simply not feasible to perform coupled polymer and reaction-diffusion simulations. We are forced to do one or the other implicitly. More importantly, not enough is known about how MukBEF interacts with DNA and other MuBEF molecules. Many of the investigations of the role played by nucleoid associated proteins such as SMC/MukBEF in organising the chromosome have been chromosome-centric in nature e.g. chromosome conformation capture (Hi-C) and related techniques. By that, we mean that they generally address the organisation of the chromosome with request to itself, not with respect to the cell. Furthermore, the fact that SMC and MukBEF form positioned foci in both *E. coli* and *B. subtilis* is not reflected in the Hi-C maps (PMID: 24158908, 29358050). In particular, though both condensins colocalise with ori, there is little evidence of an enrichment of induced contacts in this region. This disparity between the Hi-C maps and fluorescence microscopy results has yet to be explained.

Polymer modelling approaches have been used to try to understand how bacterial SMC is organising the chromosome and to explain the observed Hi-C maps (e.g. PMID 30333247). However, SMC is modelled implicitly as sliding links between nodes of the polymer (similar to studies of eukaryotic SMC e.g. PMID: 27192037). Thus, the fact that SMC form foci is not taken account of. The first mechanistic model (as in chemical kinetics) for how SMC interacts with the DNA and changes between its states (thereby translocating along the DNA and extruding loops) has very recently been published last month (PMID: 31175837) but we still do not know how the details of how it moves and self-interacts on the large spatial scale of the nucleoid.

Rather than taking a chromosome-centric approach, we take a protein-centric (and cell-centric) approach in which we are interested in the positioning of MukBEF on the bulk nucleoid. However, the price of modelling the spatial protein dynamics is that we must treat the DNA polymer implicitly. In previous work, we showed how MukBEF could self-organise on the nucleoid and be positioned by a flux-balance mechanism. In this manuscript, we provide evidence that MukBEF foci position *ori* and show how this can be explained by an interaction between ori and self-positioning MukBEF foci. We make contact with the chromosome-centric approach in the final section by performing polymer simulations to test the hypothesis that a spatially-dependent probability for ori to form contacts (loops) with other loci can lead to directed motion of ori, in a diffusion-ratchet/DNA relay -type mechanism.

In the future, we would like to shorten the gap between the two approaches. One possibility is to perform hybrid coupled simulations i.e. polymer simulations with the protein modelled implicitly coupled to reaction-diffusion simulations with the polymer modelled implicitly, using the output of one as the input to the other and vice versa in an iterative manner.

Ori, after all, is an extremely small fraction of the chromosome. How does the ori positioning explain the principal organization of the rest of the chromosome?

This is out of the scope of our paper. In our MukBEF-ori model, we study only the positioning and partitioning of ori and the chromosome is treated implicitly.

However, this question has already been investigated. There is experimental evidence that the position of the ori defines the other macrodomains and affects their position inside the cell, see PMID 28486476 and the paper cited by a reviewer in Prediction 2 below. These results are supported by polymer simulations that studied the effect of forced localisation of ori and other loci/macrodomains on overall chromosome organisation (Junier et al., 2014).

However, to address this reviewer’s concern, we have now added histograms of the positions of L, R and ter (Figure 5—figure supplement 1). We found the same organisation as in Junier et al. The mid-cell positioning of ori results in a L-ori/ter-R organisation of the polymer. Note that the polar ter localisation in Figure 3 of the aforementioned paper must be due to the histogram being from only a single simulation. By symmetry, the ter distribution in these simulations must be centred around mid-cell.

Our simulations do not take into account the formation of other macrodomains or the forced localisation of ter. These effects were considered by Junier et al., and we refer the reviewer to their paper.

The authors did add limited 3D polymeric simulations, but they are far from being sufficient to justify the main claim of the paper.

The main claim of the paper stands without the polymer simulations (in point 10, one reviewer even raises the possibility of removing them altogether). However, as evidenced by point 7, it is a natural question to ask about the mechanism underlying the biased diffusion of ori towards MukBEF foci, something which is indicated by our experimental data and which our model assumes. The goal of the polymer simulations was to propose an explanation. Even if this reviewer thinks the proposed mechanism is not plausible, the experimental data (here and previous) nonetheless indicate that ori are attracted to MukBEF foci.

4) Ori dynamics during (non-overlapping) replication.In the last part of the manuscript, the authors do consider the polymeric nature of the chromosome and model the gradient of MukBEF implicitly by dynamic looping model with only one ori. They show that the ori can have a directed motion towards the cell center where the looping probability is more, but it is not clear how the oris reach the quarter positions during replication using the dynamic looping model. How does the dynamic looping model explain the position of MukBEF at quarter positions?

We have now added substantial new polymer simulations to demonstrate ori segregation and quarter positioning. Even without looping, the ori naturally move away from one another due to entropic repulsion. However, we found that MukBEF induced looping at the quarter positions accelerates separation. The duplicated ori are then positioned at opposite quarter positions in the same way as the single ori case. See the revised text for details.

5) Ori dynamics during multifork replication.We cannot assess the robustness of preferential loading of MukBEF on ori segregation and positioning during multifork replications (when 2 or 3 replication cycles overlap with up to 8 oris). This must be one of the most straightforward predictions the authors should be able to make based on their model, and they should extend their work to multifork replication cases. To be clear, we are not asking for more experiments, just falsifiable predictions using their model for future experimental tests.

We have now examined the case of multifork replication. As it is more speculative than the slow-growing case (where we have data to constrain the model) and is somewhat out of scope of the manuscript, which is explicitly concerned with ori positioning in slow-growing cells, we have added this to the Discussion section. In short, we found very similar behaviour to the observed ori dynamics. We can use this to make predictions about the number and position of MukBEF foci in these cells. A proper investigation of multifork replication, to the same extent as we have done for non-overlapping replication, would be substantially more work and we leave it for a future manuscript.

6) Predictions.In our view, modeling would be far more informative when it makes falsifiable predictions, rather than when the ad-hoc assumptions can produce results that look like published data.

We address this comment regarding ah-hoc assumptions in the next point.

We ask the authors make two predictions, and include them in their revised manuscript.Prediction 1: what will happen to low-copy number plasmids without partitioning system, when they instead integrated the E. coli chromosome ori? We presume the authors would predict these plasmids would completely localize with MukBEF clusters given that entropic repulsion would be negligible. If validated, this prediction would significantly support both the authors MukBEF model as well as the role of entropic repulsion.

This is something we have been thinking about. We would indeed predict that they would co-localise with MukBEF foci. However, the essential point is that it is not the origin of replication site oriC itself that is important but rather one of the hypothetical loading sites, which could be some distance from oriC. For example, in *B. subtilis*, the closest parS site (outside of the ParB gene) is ~48 kb from oriC (parB is ~10kb from oriC). This is also consistent with the paper cited below in which the ori1 locus within the ori macro-domain remains positioned at mid-cell even if oriC itself is moved. Thus the challenge would be to find this site(s). It is worth noting that inserting the matS site into a plasmid leads to plasmid localisation at the septum (where the matS binding protein MatP is anchored) (PMID:22580828). So a single short sequence can indeed determine plasmid positioning much like a native partitioning system.

We have now added this prediction to the Discussion section in a separate subsection (along with the case of multi-fork replication).

Prediction 2: what would happen to the ectopic ori in the chromosome, depending on its locus position in the chromosome? And what would happen to the ectopic ori during multifork replication? Such strains were constructed previously and tested in limited growth conditions (PMID:21670292), and it would be worthwhile to extend the authors work by making explicit predictions assuming various different locations of the ectopic ori under different growth conditions.

This is unfortunately outside of the limits of our model. We do not model (apart from in the last section) the chromosome explicitly and therefore we cannot say anything about ectopic ori. Our manuscript is also concerned with non-overlapping, not multifork replication. In fact, even just simulating the polymer aspects of the question, and ignoring MukBEF, would be a massive undertaking. Indeed, we are not aware of any polymeric studies that have examined multifork replication and know only of a single paper in which active non-overlapping replication is simulated (PMID:16885211).

See also other suggested predictions above.7) The model is based on the known flux-balance mechanism and extended to explain the movement of the chromosomal origins. With each experimental observation (separation of two oris, ori dynamics during growth) they add a new aspect to the model (preferential loading, repulsive interaction of oris), which then leads to good agreement of the simulation results with the data.

Firstly, it was clear to us at the initial conception of this work that ori simply moving up the MukBEF gradient would not lead to ori partitioning. Preferential loading was then a natural effect to investigate given the role it plays for SMC, a functional homolog of MukBEF (which we now make clearer, see below). We presented the case without preferential loading first for expositional purposes as we thought that immediately jumping into a stochastic Turing model with a bidirectional interaction with a diffusing particle would be too confusing for the general reader. The presentation has now been reordered so that preferential loading is introduced earlier. With regard entropic repulsion between ori, this effect is well established and it is unsurprising that it may be included in a model of ori segregation. No other aspects or model extensions were investigated and since the two mentioned inclusions are well justified, we do not consider it a disadvantage of our model or approach that we needed to include them. We also prefer to take a minimal modelling approach in which we only add components as required to explain the data, rather than including them without justification at the beginning.

This raises two important questions: a) Are there new insights the model provides beyond the data?

The model gives the first explanation of ori positioning in *E. coli,* a long-standing question in the most studied bacterial species, and is therefore readily considered insightful. Although one of us (D.S.) has suggested in the past that MukBEF foci position ori, based on the results of perturbative experiments (described in the Introduction), it was not entirely clear that it was the case and there was no known or proposed mechanism. Furthermore, the model result that feedback from ori to MukBEF (implemented as preferential loading in our model but there are other possibilities) is necessary for ori partitioning is insightful for understanding the system and interpreting future experiments.

The concept of a coupling between a self-organising protein gradient and a particle (or genetic locus) and more generally, the idea that protein self-organisation could have a role in chromosome organisation, are also, to our knowledge, new.

Judging by the final sentence of the editor’s summary, he also agrees that our work is an important contribution and gives mechanistic insight.

b) Are the model assumptions justified? In particular, is a preferential loading ratio of six, which shows the smallest variance of the peaks, realistic?

Firstly, our model is of course effective. The chromosome is not modelled explicitly and we do not know the precise chemical kinetics of MukBEF. Therefore, even if there were experimental measurements of the preferential loading ratio and the region over which it occurs, it is not clear how that would translate into our 1D, lattice model. For this reason, we assign no meaning to, or make an interpretation of, the specific ratio of 6. The is simply a value (and an intermediate one from the point of view of the effect) that reproduces the experimental data.

The authors refer to a measured preferential loading for SMC in B. subtilis. What is the preferential loading ratio here?

In *B. subtilis*, there are population average measurement (ChIP-seq) for SMC binding to parS sites (Minnen et al. 2006, Cell Reports), where a relatively broad signal is observed around parS sites and along each chromosomal arm with a ChIP/input ratio of 2 to 4 at parS sites and 8x at oriC. The 6 most prominent parS sites (of the 8) are scattered within a region surrounding the ori (9% of the genome). Note that these ratios are much less than observed for sequence-specific DNA binding proteins. Given that in our lattice-based model preferential loading occurs only in the compartment containing the ori (1 out of 25 or 4% of the total number of compartments in a 2.5 μm cell), the ratios we investigate are not unreasonable.

Is it possible to get a measurement for MukBEF in E. coli?

There is no direct evidence in *E. coli* for preferential loading sites (hence we describe it as a hypothesis), yet MukBEF foci do co-localise with ori and, as we have shown, ori appears to be attracted to MukBEF foci rather than to mid-cell per se. The nature of the association between MukBEF and ori is a long-standing question and while we have provided further experimental evidence of the relationship and showed that the hypothesis of preferential loading can, in principle, explain the data including the observed attraction of MukBEF towards ori, we do not have direct experimental evidence for preferential loading.

*Furthermore, is the chosen repulsive interaction strength / range realistic? It might be that, although repulsion contributes to the dynamics of the origins in the* in vivo *system, it is not the main factor because it might not be strong enough.*

Indeed this might be the case but that is a big question. The importance (but not the existence) of entropic repulsion is a matter of debate and it is not a question we can answer as part of this work. However, we would like to point out that we only require entropic repulsion for initial separation of ori, after which MukBEF takes over and it is no longer necessary. This is an important point as entropic forces are expected to drop sharply after the initial unmixing.

The values for the repulsive strength and range are obtained by fitting (Figure 4—figure supplement 2). Isn't it trivial that with this fitted choice of parameters the experimental observations are reproduced?

The statement is a tautology – it is a fitting. We choose the parameters by comparing the model to the experimental observations. However, it is not trivial that we were able to obtain a successful fitting. Indeed, we showed that entropic repulsion alone, no matter how strong, cannot reproduce the observations, nor can preferential loading alone. Both entropic repulsion and preferential loading are required.

We would also like to respond more fundamentally against the implication in this point that experimental measurement and confirmation need to come before theory. Theory can and often does come first (for example in particle physics) and can provide useful insights to guide future experiments. In this case, our model and our assumption of preferential loading, or rather the assumption there exists a feedback from ori positioning to MukBEF positioning can be used to guide and/or interpret future experimental results.

8) In the model, diffusion of the origins is biased by a gradient of MukBEF. This is an important assumption of the model, which raises several questions: What is the physics behind this assumption, i.e. where does the net force come from?

The last section of the paper ‘Directed movement of ori can arise from spatially-dependent looping interactions’ was an attempt to give an explanation for the biased motion of the ori. We know that MukBEF forms loops by bridging genomically distant chromosomal sites. We also know that MukBEF is spatially localised within the cell. Finally, we know that MukBEF has a special relationship with ori. Given these facts, if we assume that MukBEF has a higher probability to form loops involving the ori than other genomic loci, then we showed that this can lead, due to the elastic fluctuations of the DNA, to migration of the ori towards the location where MukBEF/the looping probability is highest, similar to the DNA relay model of ParB-parS positioning. This kind of mechanism is also known as a diffusion ratchet, essentially a way to extract useful work from the random fluctuations. This is (one possibility for) the source of the force.

This section has been expanded with new data on segregating oris and has been improved for clarity.

What determines the strength of the bias? Can the strength be measured?

Yes, the strength is given by the slope of the velocity profile (Figure 2). We now make this explicit and give the force in Newtons.

Another assumption of the model is that only the concentration of the slow species (in blue in Figure 2A) is considered for the bias of the origins. Why are not both, the slow and the fast moving, nucleoid-bound species (in green in Figure 2A) considered for the bias? Couldn't the success of the model in explaining the data hinge to a large degree on all these assumptions?

We apologise that we did not show the profile of the fast species. It is almost homogenous and therefore does not contribute much in comparison to the gradient of the sharp peak of the slow species (Murray and Sourjik, 2017). Furthermore, we know experimentally that MukBEF foci are composed of the slowly diffusing dimer of dimer state (Badrinarayanan et al. 2012). We have modified the line in the Introduction and the Materials and methods section to make this clear.

9) The authors justify their assumption of a one-dimensional model with the fact that most of the foci are within 40% of cell width and "There are also technical reason inherent to the method that make nonlinear models in higher dimensions (and the corresponding lower voxel volumes) problematic." However, 40% of cell width is not a very confined region. Furthermore, could the authors state more clearly, which technical reasons they refer to? If it is the lower bound on the mesh size, there are generalized RDMEs that overcome this limit (Hellander and Petzold, 2016).

Please see our response to point #1.

10) Most of the parameters for the MukBEF dynamics are chosen as previously described (Murray and Sourjik, 2017). This reference should be cited when the parameters are discussed (subsection “Parameters”). The only parameter that is chosen differently is the concentration of MukBEF, which is set to a larger value than the one used before. Why is this choice justified? Couldn't the concentration be measured experimentally? Since the strength of stochastic effects depends on the number of particles, it is crucial to choose realistic concentrations / number of particles as well as realistic rates in the model.

The reason for increasing the total concentration of MukBEF was, as stated, to make splitting of the focus MukBEF occur at cells length more within the range studied in this paper (or equivalently to decrease the wavelength of the pattern). The previous paper simulated cells with lengths 3-6 μm long, whereas in this paper, we simulate cells 2.5-5 μm long (based on our data and the data of Kuwada et al.). The concentration of MukBEF in the cell has been measured experimentally by two different approaches. Single-molecule microscopy gives a value of 438+/-180 molecules of MukB (Badrinarayanan et al. 2012) in minimal media. Ribosome profiling gives values of 1215 and 356 molecules synthesised per generation for MOPS complete and minimal media respectively (Li et al. 2014). An earlier Western blot analysis gives 400 ± 100 molecules per cell (Petrushenko et al., 2006 J.Biol.Chem). Hence, the value we use (520 nM which corresponds to 391 molecules at birth (2.5 μm), is within the biologically justified range. We could alternatively have changed the parameters β and/or γ to obtain a similar result. We have now added these references and made explicit that the value is biologically justified.

11) The authors did a number of experiments with E. coli cells, in which they imaged the ori location(s) and/or Muk-foci. Together with what is already known in the literature, these experiments constitute the empirical basis for the analysis of the authors. However, instead of presenting this empirical basis in a coherent fashion in the beginning of the Results section, the authors chose to present the results piecemeal. In particular, the results that we found most striking, the "restoring velocities" of Figure 5B and D, are shown very late in the paper and presented only as a confirmation of their hypothesis of preferential Muk-loading onto the ori locus. In our view, this data should be the motivation for the hypothesis – prior to seeing this data we kept wondering what the basis for the hypothesis was.

We apologize that the motivation for preferential loading was lost to some extent during editing. The motivation was simply by analogy with SMC, which is specifically loading onto the chromosome at parS sites within the ori region. It was obvious to us from the beginning that ori moving up the MukBEF gradient would not be sufficient for partitioning and therefore preferential loading, inspired by SMC, was a reasonable hypothesis. It was only later that we examined the experimental restoring velocities of ori and MukBEF foci.

While the ori restoring velocity (panel B) supported the existing hypothesis that ori is attracted to/positioned by MukBEF foci, the MukBEF restoring velocity (panel D) was an experimentally surprising result. That MukBEF foci could be attracted to ori was not considered previously and was a prediction that came directly from the self-organising nature of the model. If MukBEF was not self-organising and was simple recruited to the ori, where it formed foci, then such a dynamic attraction of foci towards ori would not be expected.

While we think a back-and-forth between theory and experiment is a natural and truthful way to present the work, we can see how moving this data earlier might improve the overall flow of the manuscript for a broader readership. We have therefore re-organised the paper as suggested. Figure 5B and D are now part of Figure 1. However, we still refer back to Panel D as support for our hypothesis of preferential loading.

Then, after finally seeing the "restoring velocities" of Figure 5B and D, we thought that the most logical next step would be to repeat this analysis on different variants of the MukBEF-system, in order to test whether the higher restoring velocity towards MukB than to midcell is lost when the system is modified in one or the other way.

We have considered this possibility, however the only appropriate mutant that binds chromosome and forms a foci we have, the ATP hydrolysis impaired mutant mukB[E1407Q], shows very little ATP activity. We have no way to modify the activity of the system, while keeping it functional. There are several ways to break the system, but then there are either no foci or foci are no longer properly positioned within the cell (and do not colocalise with ori). Should it become possible in the future to modulate the activity of the system, then this would certainly be a very interesting approach.

Instead, the authors continued by presenting computer simulations of chromosome dynamics within a dynamic loop model. The presentation was so brief and the analysis of the simulations so minimal that the conclusions from this part of the work remained obscure to us. We think these simulations were meant to explore whether the directed motion of ori (within their model) is due to a real mechanical force on ori (and, if so, where this force comes from), or whether it results from a ratchet-like mechanism. However, what is the conclusion on this question and how do the authors arrive at the conclusion? We think this part of the work should either be eliminated or should be elaborated on and presented much more clearly.

We are somewhat surprised by these comments. In point 7, a reviewer specifically asks about the physics behind the assumption that the diffusion of ori is biased by the gradient of MukBEF. This is what the polymer simulations are trying to answer. Thus we are surprised at the suggestion to remove them entirely.

However, the section was obviously not written clearly enough. We have re-written it for clarity and also added the results of new data on the case of segregating ori.

12) The logic underlying the simulations of the stochastic model of Figure 2C was much clearer. However, given that this model relies heavily on the prior work of Murray and Sourjik, 2017, it would be good to spell out more clearly what has already been established in that paper and what is added here.

The core MukBEF model is common to this manuscript and the previous work. We go further by incorporating ori, its attraction up the MukBEF gradient, its effect (via preferential loading) on MukBEF foci and the repulsion between ori and how all these things together give rise robust ori positioning and partitioning. We have edited the Discussion to make it clearer what was previously achieved and what we do in this manuscript (this is already commented on in the Introduction).

Furthermore, the way in which the comparison of these simulations with the experimental data is done is sometimes confusing, in the sense that we did not get a clear sense about "what is put in and what they got out". For instance, it says "… Note this was based on a direct comparison and not a fit to the experimental distribution.", while earlier, it says "… We found that by adjusting the diffusion and drift input parameters, we were able to obtain excellent agreement with the experimental results (Figure 2F)." (the latter indicates fitting).

We have made this clearer by changing the order of presentation and stating the motivation for the preferential loading more clearly. The model was not fit to all existing data. We determined the parameters defining the ori diffusion constant and drift rate by fitting to the mean and variance of the velocity profile (Figure 2F). We then compared (not fitted) other properties of the model with other experimental data, namely the distribution of the ori-MukBEF separation distance, where we found excellent agreement (in the presence of specific loading). As this information was not used to constrain the model, it can be seen as confirmation of a model prediction. This was our point here. We have made this clearer.